# FLOW MATCHING FOR ACCELERATED SIMULATION OF ATOMIC TRANSPORT IN MATERIALS

## ABSTRACT

We introduce LIFLOW, a generative framework to accelerate molecular dynamics (MD) simulations for crystalline materials that formulates the task as conditional generation of atomic displacements. The model uses flow matching, with a *Propagator* submodel to generate atomic displacements and a *Corrector* to locally correct unphysical geometries, and incorporates an adaptive prior based on the Maxwell–Boltzmann distribution to account for chemical and thermal conditions. We benchmark LIFLOW on a dataset comprising 25-ps trajectories of lithium diffusion across 4,186 solid-state electrolyte (SSE) candidates at four temperatures. The model obtains a consistent Spearman rank correlation of 0.7–0.8 for lithium mean squared displacement (MSD) predictions on unseen compositions. Furthermore, LIFLOW generalizes from short training trajectories to larger supercells and longer simulations while maintaining high accuracy. With speed-ups of up to $600,000\times$ compared to first-principles methods, LIFLOW enables scalable simulations at significantly larger length and time scales.

## 1 INTRODUCTION

Atomic transport is a fundamental process that governs the performance of materials in various technologies, including energy storage, catalysis, and electronic devices (Balluffi et al., 2005; Yip, 2023). Solid-state electrolytes (SSEs) are a prime example, emerging as a safer and more stable alternative to liquid electrolytes commonly used in lithium-ion batteries (Bachman et al., 2016). The study and design of SSEs rely on fast and accurate atomistic simulation techniques to model the intricate ionic diffusion behaviors that dictate the atomic transport in these materials. The standard method, *ab initio* molecular dynamics (AIMD), involves costly density functional theory (DFT) calculations for each propagation step in the scale of femtoseconds. Hence, their application is limited to small spatiotemporal scales and a few simulations, often insufficient for characterizing diffusive dynamics or screening candidate materials. Recently, universal machine learning interatomic potentials (MLIPs), trained on large-scale DFT calculations, have emerged as a promising alternative (Friederich et al., 2021; Ko & Ong, 2023). However, even with MLIPs, dynamics must be discretized in sufficiently small time steps to ensure stable and accurate propagation (Fu et al., 2023a) and are still too slow to enable scalable simulation to perform high-throughput screening from large material databases.

To accelerate MD simulations for small bio/organic molecules, methods such as Timewarp (Klein et al., 2023a), Implicit Transfer Operator Learning (ITO, Schreiner et al. (2023)), Score Dynamics (SD, Hsu et al. (2024)), and Force-Guided Bridge Matching (FBM, Yu et al. (2024)) have been proposed. These methods leverage a generative model to propagate the conformational distribution from time $\tau$ to time $\tau + \Delta\tau$, where $\Delta\tau$ is much larger than the typical MD time steps. A similar approach has been applied to coarse-grained polymer electrolyte simulations in Fu et al. (2023b).

In the context of all-atom simulations of crystalline materials across different temperatures, these methods do not account for symmetries or handle various atom types under various simulation conditions. This work aims to address this by developing a tailored, flow-matching-based, generative acceleration framework designed for scalable and cost-effective simulations of crystalline materials and diffusive dynamics in SSEs. The key objective is to construct a model capable of accurately reproducing relevant kinetic observables, such as mean squared displacement (MSD) and self-diffusivity of mobile ions, in comparison to long MD simulations using MLIPs or AIMD.

Our contributions are as follows:

- We introduce the task of generative acceleration of MD for crystalline materials, formulating it as the conditional generation of atomic displacements. Our approach accounts for periodic boundary conditions and generalizes effectively across different supercell sizes.

- We develop a flow matching approach with a physically motivated adaptive prior to account for chemical and thermal conditions, along with a corrector mechanism to ensure stability.

- Additionally, we contribute a trajectory dataset based on MLIPs, designed to benchmark performance across diverse material systems and temperature conditions.

## 2 BACKGROUND

### 2.1 PRELIMINARIES

**Crystalline materials and representation** The crystal structure, assuming perfect order with translational symmetry, can be idealized as an infinite repetition of atoms, each assigned an atom type from the periodic table $\mathcal{A}$, within a unit cell with periodic boundary conditions (Ashcroft & Mermin, 1976). In this work, the structure of a material with $n$ atoms in the unit cell is represented by the tuple $\mathcal{M} = (\boldsymbol{X}, \boldsymbol{L}, \boldsymbol{a})$, where $\boldsymbol{X} = (\boldsymbol{x}_1, \boldsymbol{x}_2, \cdots, \boldsymbol{x}_n)^\top \in \mathbb{R}^{n \times 3}$ denotes the Cartesian coordinates of the atoms, $\boldsymbol{L} = (\boldsymbol{l}_1, \boldsymbol{l}_2, \boldsymbol{l}_3)^\top \in \mathbb{R}^{3 \times 3}$ with rows defining the basis vectors of a 3-D repeating unit cell, and $\boldsymbol{a} \in \mathcal{A}^n$ is the atom types. We impose a graph structure on the material by connecting pairs of nearby atoms with edges (Schütt et al., 2017), possibly across unit cell boundaries. An edge $((i, j), \boldsymbol{k}) \in [\![1, n]\!]^2 \times \mathbb{Z}^3$ is formed between atoms $i$ and $j$ if the distance between atom $i$ and atom $j$, displaced by $\boldsymbol{k}$ unit cells from $i$, is smaller than the cutoff, i.e., $\|\boldsymbol{x}_j + \boldsymbol{k}\boldsymbol{L} - \boldsymbol{x}_i\|_2 < r_{\text{cutoff}}$. An $a \times b \times c$ supercell of $\mathcal{M}$ is defined as

$$(\boldsymbol{X}', \boldsymbol{L}', \boldsymbol{a}') = (\oplus_{\kappa=1}^{abc}(\boldsymbol{X} + \boldsymbol{1}_n \otimes \boldsymbol{k}_\kappa), \boldsymbol{L}\operatorname{diag}(a, b, c), \oplus_{\kappa=1}^{abc}\boldsymbol{a}), \tag{1}$$

where $\oplus$ denotes concatenation, $\otimes$ is the outer product, and $\boldsymbol{k}_\kappa \in \mathbb{Z}_a \times \mathbb{Z}_b \times \mathbb{Z}_c$ represents the index of unit cell repetitions. Although the method is designed for general crystalline materials, the primary application of this work is on lithium SSEs, which are further discussed in Appendix A.1.

**Molecular dynamics for materials** MD is a simulation methodology used to sample ensembles of configurations or trajectories of atomistic systems by solving the equation of motion, $\boldsymbol{M}\ddot{\boldsymbol{X}}_\tau = -\nabla U(\boldsymbol{X}_\tau)$, over time $\tau$,[1] where $\boldsymbol{M} = \operatorname{diag}(\boldsymbol{m})$ is the diagonal matrix of atomic masses, $\boldsymbol{m} = \boldsymbol{m}(\boldsymbol{a})$, and $U(\boldsymbol{X})$ is the potential energy (Frenkel & Smit, 2023). Given the initial position $\boldsymbol{X}_0$ and the velocity $\dot{\boldsymbol{X}}_0$, typically sampled from the Maxwell–Boltzmann distribution, the equation of motion is usually discretized and propagated with a time step of 0.5–2 fs, depending on the fastest motion of the system. For organic and biomolecules with a limited number of atom types, classical force fields with specified functional forms are routinely used to approximate the system energy with reasonable accuracy. However, for inorganic solid systems containing various elements across the periodic table, the parametrization of classical force fields is challenging, and $U(\cdot)$ is often derived from quantum mechanical calculations (*ab initio* MD), which are computationally much more expensive and scale poorly ($\mathcal{O}(n^3)$ in theory). For more details, refer to Appendix A.2.

**Machine learning interatomic potentials** Due to the high computational cost of *ab initio* calculations, machine learning interatomic potentials (MLIPs) based on graph neural networks have been developed to approximate the results of the quantum calculations (Friederich et al., 2021; Ko & Ong, 2023). Recent advances in universal MLIPs, such as MACE-MP-0 (Batatia et al., 2024) and CHGNet (Deng et al., 2023), enable faster simulations and linear scaling with respect to number of atoms, but pre-trained models often tend to overestimate the kinetic properties (Deng et al., 2024). We employed MLIPs to generate our trajectory dataset of lithium-containing materials to demonstrate chemical transferability across different materials.

**Flow matching** Flow matching (Lipman et al., 2023) is a generative modeling framework in which samples from the prior distribution $x_0 \sim p_0(x)$ are transported to samples from the data distribution $x_1 \sim q(x)$ by a time-dependent vector field $u_t(x)$ ($t \in [0, 1]$). The vector field generates a flow $\psi_t$

---

[1]In this work, we denote physical time by $\tau$ and flow matching time by $t$. For clarity, we omit the physical time when it does not cause ambiguity, e.g., $\boldsymbol{D}_0$ and $\boldsymbol{D}_1$ correspond to $t = 0$ and 1, respectively.

defined with $\psi_0(x) = x$ and $(d/dt)\psi_t(x) = u_t(\psi_t(x))$ and a probability path $p_t(x) = [\psi_t]_* p_0(x)$. The data conditional vector field $u_t(x|x_1)$ is available in closed form for the commonly used Gaussian probability path, i.e., $p_t(x|x_1) = \mathcal{N}(x; \mu_t(x_1), \sigma_t(x_1)^2 \boldsymbol{I})$. The marginal vector field model $v_t(x; \theta)$ is parametrized by a neural network and learned by the following regression objective:

$$\mathcal{L}_{\text{CFM}}(\theta) = \mathbb{E}_{t \sim \mathcal{U}(t;0,1), x_1 \sim p_1(x), x \sim p_t(x|x_1)} \|v_t(x; \theta) - u_t(x|x_1)\|^2. \qquad (2)$$

## 2.2 RELATED WORKS

**ML surrogates for dynamics simulation**   Several works have explored ML surrogates for time-coarsened dynamics by learning transition probability densities. Timewarp (Klein et al., 2023a) employs a conditional normalizing flow (CNF) with Markov chain Monte Carlo sampling, while ITO (Schreiner et al., 2023) is a conditional diffusion model designed as an arbitrary time-lag propagator. Arts et al. (2023) models coarse-grained (CG) dynamics with diffusion models, SD (Hsu et al., 2024) learns the score function of the transition density, $F^3$low (Li et al., 2024) models protein CG frame transitions with flow matching, and FBM (Yu et al., 2024) uses a conditional bridge process with a correction mechanism based on intermediate force fields. These methods are applied to biomolecular simulations, with less chemical diversity and different symmetry requirements and task formulations from our work. Notably, Fu et al. (2023b) targets non-Markovian dynamics in CG polymer materials by learning the acceleration and using a score-based corrector. While CG allows for the explicit modeling of dynamics over longer timesteps using the equations of motion, our task requires all-atom modeling, necessitating a generative surrogate for the dynamics. Moreover, none of these approaches considered a task-specific, physically motivated adaptive prior, which was crucial to the improved performance in this work.

**Generative models for materials**   As a time-hopping conditional generative model for material structures, our approach shares design principles with crystal generation models (Xie et al., 2022; Jiao et al., 2023; AI4Science et al., 2023; Zeni et al., 2024; Yang et al., 2024; Miller et al., 2024), which use diffusion or flow matching to generate atomic identities and positions within a unit cell. While these methods often handle position generation as fractional coordinates with periodic boundaries, our task requires modeling displacements in Cartesian coordinates directly without wrapping positions back into the unit cell (see Appendix A.3).

**Prior design**   While the normal distribution is commonly used in diffusion and flow-based generative models, incorporating task-specific inductive biases into the prior can improve the performance. Lee et al. (2022) introduced data-dependent priors in diffusion models, Guan et al. (2023) used decomposed priors for ligand generation, Jing et al. (2023) applied harmonic priors for protein structure, and Irwin et al. (2024) employed scale-based priors for molecular conformation. The common goal in these methods is to reduce the transport cost by initializing the prior closer to the data distribution. We use a physically motivated prior based on the Maxwell–Boltzmann distribution, which additionally accounts for differences between atom types and reflects thermal and phase conditions.

# 3 METHODS

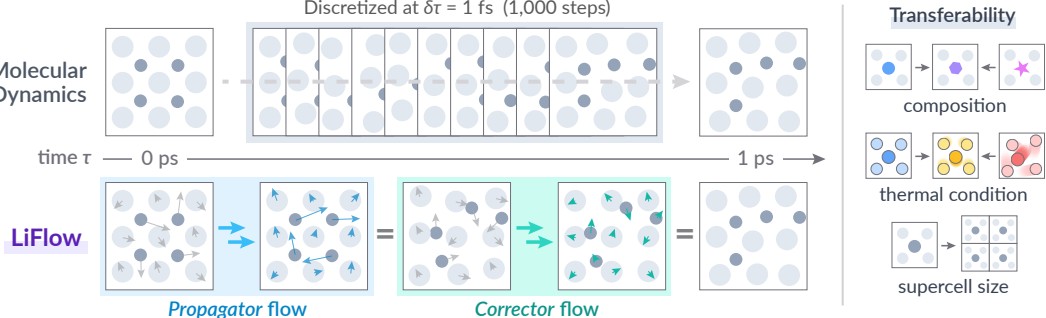

Figure 1: **LIFLOW scheme.** LIFLOW is a generative acceleration framework for MD simulations for crystalline materials, with *Propagator* and *Corrector* components leveraging a conditional flow matching scheme for accurate generation of atomic displacements during time propagation.

The LiFlow framework is illustrated in Fig. 1. We begin by outlining the problem of generating atomic displacements to accelerate simulations and discussing the symmetry constraints necessary for scalable generation. Next, we propose a physically motivated prior and a flow model parametrization that adheres to these constraints, followed by the training and inference processes.

## 3.1 PROBLEM SETTING

Similar to the ML-based MD acceleration methods in Section 2.2, our goal is to model the transition probability density of a material structure over a time interval $\Delta\tau$, conditioned on the temperature $T$: $p(\mathcal{M}_{\tau+\Delta\tau}|\mathcal{M}_{\tau}, T)$. For this task, we fix the lattice $\boldsymbol{L}$ (constant volume) and atom types $\boldsymbol{a}$, and set $\Delta\tau$ as 1 ps, which is 1,000 times larger than the usual MD time step of 1 fs (Marx & Hutter, 2009). In MD simulations used to model the kinetics of materials, *unwrapped* coordinates are utilized, meaning atomic coordinates are not confined to the unit cell, in order to keep track of the atomic displacements (von Bülow et al., 2020). As a result, unlike previous ML surrogates for dynamics of bio/organic molecules (Section 2.2) with a single connected component with the fixed center of mass, the distribution of positions does not have a finite support. Therefore, we opt to model the distribution of *displacements* over time interval $\Delta\tau$, $\boldsymbol{D}_{\Delta\tau} := \boldsymbol{X}_{\tau+\Delta\tau} - \boldsymbol{X}_{\tau}$. In summary,

> **Task:** learn the conditional distribution of atomic displacements $p(\boldsymbol{D}_{\Delta\tau}|\boldsymbol{X}_{\tau}, \boldsymbol{L}, \boldsymbol{a}, T)$ from a dataset of time-separated pairs of structures $\mathcal{D} = \{((\boldsymbol{X}_{\tau}, \boldsymbol{X}_{\tau+\Delta\tau}), \boldsymbol{L}, \boldsymbol{a}, T))\}$, extracted from MD trajectories across various material compositions and temperatures.

More details and rationale on the task design choices can be found in Appendix A.3.

## 3.2 CONDITIONAL FLOW MATCHING FOR TIME PROPAGATION

### 3.2.1 SYMMETRY CONSIDERATIONS

The conditional probability density of displacements is invariant to permutation of atomic indices, global translation and lattice shift of atomic coordinates, global rotation applied to relevant variables, and supercell choice (we omit the physical time $\tau$ and $\Delta\tau$ here for brevity):

$$p(\boldsymbol{D}|\boldsymbol{X}, \boldsymbol{L}, \boldsymbol{a}, T) = p(\boldsymbol{P}\boldsymbol{D}|\boldsymbol{P}\boldsymbol{X}, \boldsymbol{L}, \boldsymbol{P}\boldsymbol{a}, T), \qquad \boldsymbol{P} \in S_n \text{ (permutation)} \quad (3)$$

$$p(\boldsymbol{D}|\boldsymbol{X}, \boldsymbol{L}, \boldsymbol{a}, T) = p(\boldsymbol{D}|\boldsymbol{X} + \boldsymbol{1}_n \otimes \boldsymbol{t}, \boldsymbol{L}, \boldsymbol{a}, T), \qquad \boldsymbol{t} \in \mathbb{R}^3 \text{ (global translation)} \quad (4)$$

$$p(\boldsymbol{D}|\boldsymbol{X}, \boldsymbol{L}, \boldsymbol{a}, T) = p(\boldsymbol{D}|\boldsymbol{X} + \boldsymbol{Z}\boldsymbol{L}, \boldsymbol{L}, \boldsymbol{a}, T), \qquad \boldsymbol{Z} \in \mathbb{Z}^{n \times 3} \text{ (lattice periodicity)} \quad (5)$$

$$p(\boldsymbol{D}|\boldsymbol{X}, \boldsymbol{L}, \boldsymbol{a}, T) = p(\boldsymbol{D}\boldsymbol{R}|\boldsymbol{X}\boldsymbol{R}, \boldsymbol{L}\boldsymbol{R}, \boldsymbol{a}, T), \qquad \boldsymbol{R} \in \mathrm{O}(3) \text{ (rotation/reflection)} \quad (6)$$

$$p(\boldsymbol{D}|\boldsymbol{X}, \boldsymbol{L}, \boldsymbol{a}, T) = p(\boldsymbol{D}'|\boldsymbol{X}', \boldsymbol{L}', \boldsymbol{a}', T), \qquad \text{(supercell, defined as Eq. (1))} \quad (7)$$

In general, to model the invariant densities with CNFs, we need an invariant base distribution and equivariant flow vector fields (Köhler et al., 2020; Klein et al., 2023b). Translational invariances Eqs. (4) and (5) and supercell invariance Eq. (7) are satisfied by our choice of representation for materials (Section 2.1). For O(3) and $S_n$ symmetries, we model our prior and flow according to the following proposition.

**Proposition 1** *Given an invariant base distribution $p_0(\boldsymbol{D}_0)$ satisfying Eqs. (3) and (6) and an equivariant conditional vector field $u_t(\boldsymbol{D}_t|\boldsymbol{D}_1)$ with the following properties:*

$$u_t(\boldsymbol{P}\boldsymbol{D}_t|\boldsymbol{P}\boldsymbol{D}_1, \boldsymbol{P}\boldsymbol{X}, \boldsymbol{L}, \boldsymbol{P}\boldsymbol{a}, T) = \boldsymbol{P}u_t(\boldsymbol{D}_t|\boldsymbol{D}_1, \boldsymbol{X}, \boldsymbol{L}, \boldsymbol{a}, T), \qquad \boldsymbol{P} \in S_n \quad (8)$$

$$u_t(\boldsymbol{D}_t\boldsymbol{R}|\boldsymbol{D}_1\boldsymbol{R}, \boldsymbol{X}\boldsymbol{R}, \boldsymbol{L}\boldsymbol{R}, \boldsymbol{a}, T) = u_t(\boldsymbol{D}_t|\boldsymbol{D}_1, \boldsymbol{X}, \boldsymbol{L}, \boldsymbol{a}, T)\boldsymbol{R}, \qquad \boldsymbol{R} \in \mathrm{O}(3) \quad (9)$$

*the generated conditional probability path $p_{t|1}(\boldsymbol{D}_t|\boldsymbol{D}_1)$ is invariant. Furthermore, given that the data distribution $q(\boldsymbol{D}_1)$ is invariant, the marginal probability path $p_t(\boldsymbol{D}_t)$ is also invariant.*

Note that the group actions of $S_n$ and O(3) on the optional conditional variable $\boldsymbol{D}_1$ are the same as their actions on $\boldsymbol{D}_t$. The proof is given in Appendix B.

### 3.2.2 CHOICE OF PRIOR

We consider a Gaussian prior, $\boldsymbol{D}_0 \sim \mathcal{N}(\boldsymbol{D}_0; \boldsymbol{0}, \boldsymbol{\Sigma} \otimes \boldsymbol{I}_3)$, with a diagonal covariance $\boldsymbol{\Sigma} = \mathrm{diag}(\boldsymbol{\sigma})^2$, where $\boldsymbol{\sigma} = \boldsymbol{\sigma}(\boldsymbol{a}, T) \in \mathbb{R}^n$ is equivariant to atom index permutation. This prior distribution satisfies

the symmetry constraints Eqs. (3) to (7). In MD simulation of materials, atomic displacements tend to be larger for lighter atoms and at higher temperatures. In the short-time, non-interacting limit, the displacements can be expressed as $\boldsymbol{D}_{\delta\tau} = \dot{\boldsymbol{X}}_\tau \delta\tau$, where the marginal distribution of velocity follows the Maxwell–Boltzmann distribution, $\dot{\boldsymbol{X}}_\tau \sim \mathcal{N}(\dot{\boldsymbol{X}}_\tau; \boldsymbol{0}, \mathrm{diag}(k_{\mathrm{B}} T / \boldsymbol{m}) \otimes \boldsymbol{I}_3)$, with $k_{\mathrm{B}}$ being the Boltzmann constant. Thus, it is reasonable to initialize the noise from a *scaled* Maxwell–Boltzmann distribution with $\boldsymbol{\sigma} = \sigma \cdot (k_{\mathrm{B}} T / \boldsymbol{m})^{1/2}$, where $\sigma$ is a constant hyperparameter controlling the scale.

In the specific context of AIMD simulations in this work, where the simulations are often conducted at elevated temperatures, the material may undergo phase transitions (e.g., from solid to liquid) within the temperature range covered by the dataset. Additionally, for lithium-based solid-state electrolytes, lithium atoms may exhibit displacements several orders of magnitude larger than those of non-lithium (frame) atoms. To account for these variations, we introduce a material-dependent *adaptive* scaling factor for the Maxwell–Boltzmann distribution:

$$\boldsymbol{\sigma} = [\sigma_{\mathrm{Li}}(\mathcal{M}_0, T) \cdot \mathbb{I}_{\boldsymbol{a}=\mathrm{Li}} + \sigma_{\mathrm{frame}}(\mathcal{M}_0, T) \cdot \mathbb{I}_{\boldsymbol{a}\neq\mathrm{Li}}] \odot (k_{\mathrm{B}} T / \boldsymbol{m})^{1/2}, \tag{10}$$

where for each species $\mathcal{S} \in \{\text{lithium}, \text{frame}\}$, $\sigma_{\mathcal{S}}$ selects a scale value from the hyperparameters $\{\sigma_{\mathcal{S}}^{\mathrm{small}}, \sigma_{\mathcal{S}}^{\mathrm{large}}\}$ based on a binary classifier's prediction of whether the displacements for $\mathcal{S}$ will be small or large. The classifier utilizes temperature and the average-pooled atomic invariant features of $\mathcal{S}$, extracted from a pre-trained MACE model, based on the initial material structure $\mathcal{M}_0$. Further details about the classifier model are provided in Appendix D.1.

### 3.2.3 FLOW PARAMETRIZATION

**Conditional flow matching** Following Pooladian et al. (2023), we select the linear interpolation between the prior sample and the data sample as a conditional flow:

$$u_t(\boldsymbol{D}_t | \boldsymbol{D}_1) = \frac{\boldsymbol{D}_1 - \boldsymbol{D}_t}{1 - t} \quad \text{and} \quad \boldsymbol{D}_t = \psi_t(\boldsymbol{D}_0 | \boldsymbol{D}_1) = (1 - t)\boldsymbol{D}_0 + t\boldsymbol{D}_1. \tag{11}$$

This satisfies the symmetry constraints in Eqs. (8) and (9). The marginal flow approximator $v_t(\boldsymbol{D}_t, \boldsymbol{X}_\tau, \boldsymbol{L}, \boldsymbol{a}, T; \theta)$ should also respect these symmetry constraints. We adopt the PAINN model (Schütt et al., 2021) to balance expressiveness with inference speed. PAINN is an equivariant graph neural network that outputs scalar and vector quantities based on the atomistic graph, incorporating scalar and vector node features. The structure is encoded using a radial basis function expansion of atomic distances and the unit vector directions along edges (Eqs. (26a) and (27a)). We observed that encoding the intermediate structure $\boldsymbol{X}_\tau + \boldsymbol{D}_t$ significantly improves prediction performance (Table A1). Thus, we modify the message-passing layers of PAINN to accept two structural inputs: $\boldsymbol{X}_\tau$ and $\boldsymbol{X}_\tau + \boldsymbol{D}_t$. Additionally, the intermediate displacements $\boldsymbol{D}_t$ are used to construct the vector node features. Further details on the model architecture and modifications are given in Appendix D.2.

***Propagator* and *Corrector* models** While, in theory, a single generative model should suffice to learn the density, prediction errors would arise from two sources: inaccuracies in the marginal flow prediction and discretization errors in the flow integration. Moreover, since the trajectory generation is performed autoregressively, applying the generative model iteratively compounds these errors over time. To address this, in addition to the flow matching model described earlier (*Propagator*), we introduce an auxiliary flow matching model named *Corrector*, inspired by Fu et al. (2023b), to rectify potential errors in the predicted displacements.

Although the *Corrector* model is intended to correct errors in the final displacement resulting from the integration of *Propagator*, directly mapping the generated output to an actual data sample to compute the target correction value can be complex, as it may require differentiating through the flow integration. Therefore, we decouple the *Propagator* and *Corrector* models, training the *Corrector* to denoise positional noise of arbitrary small scale. Given a perturbed configuration $\tilde{\boldsymbol{X}}_\tau = \boldsymbol{X}_\tau + \boldsymbol{D}$, where the noise displacement is sampled from $\boldsymbol{D} | \boldsymbol{\sigma}' \sim \mathcal{N}(\boldsymbol{D}; \boldsymbol{0}, \mathrm{diag}(\boldsymbol{\sigma}')^2 \otimes \boldsymbol{I}_3)$ with the noise scale $\boldsymbol{\sigma}' \sim \mathcal{U}(\boldsymbol{\sigma}'; \boldsymbol{0}, \sigma_{\mathrm{max}} \boldsymbol{1}_n)$, the flow is trained to generate the possible denoising displacements $-\boldsymbol{D}$ conditioned on $\tilde{\boldsymbol{X}}_\tau$.

### 3.2.4 TRAINING AND INFERENCE

**LIFLOW training** We train the model using time-separated pairs of structures, $((\boldsymbol{X}_\tau, \boldsymbol{X}_{\tau'}), \boldsymbol{L}, \boldsymbol{a}, T))$, sampled from MD trajectories in the training set. First, the prior displacements are sampled based on the possible choices outlined in Section 3.2.2. The *Propagator* and *Corrector* are trained to approximate the marginal flows toward the distributions of the possible propagating displacements, $\boldsymbol{X}_{\tau'} - \boldsymbol{X}_\tau$, and denoising displacements, $\boldsymbol{X}_{\tau'} - \tilde{\boldsymbol{X}}_{\tau'}$, respectively. These are conditioned on the previous structure, $\boldsymbol{X}_\tau$, and the noisy structure, $\tilde{\boldsymbol{X}}_{\tau'}$, respectively. Given interpolated displacements $\boldsymbol{D}_t$ and the corresponding conditional variables, both models are trained to match the ground truth conditional flow $u_t(\boldsymbol{D}_t|\boldsymbol{D}_1)$ using the regression loss Eq. (2). Detailed training algorithms are reported in Appendix D.3.

---

**Algorithm 1:** LIFLOW Inference

**Input:** Initial position $\boldsymbol{X}_0$, lattice $\boldsymbol{L}$, atom types $\boldsymbol{a}$, atomic masses $\boldsymbol{m}(\boldsymbol{a})$, temperature $T$

**Output:** Predicted position $\boldsymbol{X}_\tau$ at $\tau = N_{\text{step}}\Delta\tau$

Determine the prior from $\boldsymbol{X}_0$, $\boldsymbol{L}$, $\boldsymbol{a}$, $\boldsymbol{m}$, and $T$

**for** $i_\tau \leftarrow 0$ **to** $N_{\text{step}} - 1$ **do**
    $\tau \leftarrow i_\tau\Delta\tau$ and $\tau' \leftarrow (i_\tau + 1)\Delta\tau$
    Sample $\boldsymbol{D}$ from the *Propagator* prior
    **for** $i \leftarrow 0$ **to** $N_{\text{flow}} - 1$ **do**
        $\boldsymbol{D} \leftarrow \boldsymbol{D} + Propagator(\boldsymbol{D}, \boldsymbol{X}_\tau, \boldsymbol{L}, \boldsymbol{a}, T, t)/N_{\text{flow}}$
    $\tilde{\boldsymbol{X}}_{\tau'} \leftarrow \boldsymbol{X}_\tau + \boldsymbol{D}$      // Propagator step
    Sample $\boldsymbol{D}$ from the *Corrector* prior
    **for** $i \leftarrow 0$ **to** $N_{\text{flow}} - 1$ **do**
        $\boldsymbol{D} \leftarrow \boldsymbol{D} + Corrector(\boldsymbol{D}, \tilde{\boldsymbol{X}}_{\tau'}, \boldsymbol{L}, \boldsymbol{a}, T, t)/N_{\text{flow}}$
    $\boldsymbol{X}_{\tau'} \leftarrow \tilde{\boldsymbol{X}}_{\tau'} + \boldsymbol{D}$      // Corrector step
    $\boldsymbol{X}_{\tau'} \leftarrow \boldsymbol{X}_{\tau'} - \text{CoM}(\boldsymbol{X}_{\tau'}, \boldsymbol{m}) + \text{CoM}(\boldsymbol{X}_\tau, \boldsymbol{m})$

---

**LIFLOW inference** The inference procedure is provided in Algorithm 1. Starting from the initial atom positions $\boldsymbol{X}_0$, we alternate between *Propagator* and *Corrector* flow integration for $N_{\text{step}}$ steps, generating the trajectory $\{\boldsymbol{X}_0, \boldsymbol{X}_{\Delta\tau}, \cdots, \boldsymbol{X}_{N_{\text{step}}\Delta\tau}\}$. The flow integration for both the *Propagator* and *Corrector* begins by sampling prior displacements $\boldsymbol{D}_0$ from the chosen prior distribution. These displacements are then updated over $N_{\text{flow}}$ steps using Euler's method, based on the predicted marginal flow. Since MD simulations are often performed with a fixed center-of-mass (CoM) position, defined as $\text{CoM}(\boldsymbol{X}, \boldsymbol{m}) = \sum_j m_j\boldsymbol{x}_j / \sum_j m_j$, we correct for any CoM drift after each *Propagator*–*Corrector* inference step.

## 4 EXPERIMENTS

### 4.1 DATASETS AND METRICS

**Universal MLIP dataset (Section 4.2)** To train a compositionally transferable generative model for time-shifting conformational distributions, long-time simulation trajectories that span a diverse range of compositional spaces in solid-state materials are required. A total of 4,186 stable lithium-containing structures were retrieved from the Materials Project database (Jain et al., 2013) to capture various modes of lithium-ion dynamics across different compositions. For each structure, 25 ps MD simulations were performed using the MACE-MP-0 small universal MLIP model (Batatia et al., 2024) at temperatures of 600, 800, 1000, and 1200 K, with a time step of 1 fs (25k steps per structure). The distribution of elements in these structures, shown in Fig. A1a, spans 77 elements across the periodic table. The mean squared displacement (MSD) of lithium atoms for each structure over the 25 ps trajectories is shown in Fig. A1b, indicating that the dataset captures a broad range of atomic environments and dynamic behaviors. The dataset is divided into training (90%) and test (10%) sets based on material composition, with the validation set sampled from the training portion. Details of the simulations and dataset statistics are provided in Appendix C.1.

**AIMD datasets (Section 4.3)** To evaluate the ability to extend accurate atomistic dynamics from short AIMD simulations, we employed two sets of AIMD trajectories that exhibit diffusive lithium dynamics. The first set includes LPS ($Li_3PS_4$) simulations from Jun et al. (2024b), with ~250 ps trajectories for 128-atom structures of $\alpha$-, $\beta$-, and $\gamma$-LPS, conducted at 600–800 K. The second set comprises LGPS ($Li_{10}GeP_2S_{12}$) simulations from López et al. (2024), which includes ~150 ps MD trajectories for a $2 \times 2 \times 1$ supercell (200 atoms) of LGPS at temperatures of 650, 900, 1150, and 1400 K. We used the first 25 ps of each trajectory as the training set. Refer to Appendix C.2 for the rationale behind system selection and further details.

**Metrics** To quantify the prediction of kinetic observables, we compared the MSD of lithium and frame atoms between generated and reference trajectories. The MSD measures the average squared distance that particles of type $\mathcal{S}$ move over time $\tau$, as defined in Eq. (12):

$$\text{MSD}_\mathcal{S}(\tau) = \frac{1}{|\mathcal{S}|} \sum_{i \in \mathcal{S}} \|\boldsymbol{x}_{\tau,i} - \boldsymbol{x}_{0,i}\|^2 \quad (12) \qquad\qquad D_\mathcal{S}^* = \lim_{\tau \to \infty} \frac{\text{MSD}_\mathcal{S}(\tau)}{6\tau} \quad (13)$$

Given the wide range of magnitudes of MSD values, we compared the log values (base 10) of MSD, with MSD in units of $\text{Å}^2$. We report the mean absolute error (MAE) and Spearman's rank correlation ($\rho$) for the log MSD predictions on the universal MLIP dataset.

In the long-time limit, the MSD grows linearly with time, with a rate proportional to the self-diffusivity $D_\mathcal{S}^*$ (Eq. (13)). This is quantified using Bayesian regression of MSD against time (McCluskey et al., 2024a,b). According to the Arrhenius relationship, the temperature dependence of diffusivity follows $\log D^*(T) = \log D_0^* - E_A/k_B T$, where $E_A$ is the activation energy. Since activation energy is a key measure of the barrier to lithium diffusion in materials science literature, we also verify whether $E_A$ is accurately reproduced in the LGPS AIMD dataset.

To evaluate the reproduction of structural features, we compare the all-particle radial distribution function (RDF), $g(r)$. The RDF describes how particle density varies as a function of distance from a reference particle, revealing spatial organization and local structure in the system. It is defined as:

$$g(r) = \frac{1}{4\pi r^2} \frac{1}{\rho n} \sum_i \sum_{j \neq i} \delta(r - \|\boldsymbol{x}_i - \boldsymbol{x}_j\|), \quad (14)$$

where $\rho$ is the number density of atoms. We average the RDF over the latter parts of the simulation, after discarding a short induction period (5 ps, 20 % of the trajectory). The accuracy is quantified by the RDF MAE $= (1/r_\text{cut}) \int_0^{r_\text{cut}} |\hat{g}(r) - g(r)|\, dr$, with $r_\text{cut} = 5$ Å. Note that a similar set of metrics has been adopted in the benchmark of MLIP-based simulations (Fu et al., 2023a).

## 4.2 Universal Model

### 4.2.1 Setup and Results

Table 1: **Results for the universal model.** Evaluation metrics for different *Propagator* priors (isotropic and uniform/adaptive scale Maxwell–Boltzmann) with or without the *Corrector*. *Regressor*[†]: non-generative, directly predicting displacements. $P_\text{adaptive} + C$ * represents the baseline model without any ablations. Values are colored from worst to best for each metric and $T$, and standard deviations are reported in Table A4.

| Train $T$ (K) | Inference $T$ (K) | Model | log MSD$_\text{Li}$ MAE ($\downarrow$) | log MSD$_\text{Li}$ $\rho$ ($\uparrow$) | log MSD$_\text{frame}$ MAE ($\downarrow$) | RDF MAE ($\downarrow$) | Stable traj. % ($\uparrow$) |
|---|---|---|---|---|---|---|---|
| **Exp 1** | Single temperature: is Maxwell–Boltzmann prior required? | | | | | | |
| 800 | 800 | *Regressor*[†] | 1.636 | 0.535 | 0.876 | 0.416 | 90.2 |
| | | $P_\text{isotropic}$ | 0.498 | 0.753 | 0.318 | 0.113 | 98.6 |
| | | $P_\text{uniform}$ | **0.396** | **0.779** | **0.274** | **0.084** | **99.4** |
| **Exp 2** | Multiple temperatures: are adaptive prior scaling Eq. (10) and *Corrector* required? | | | | | | |
| All | 600 | $P_\text{uniform}$ | **0.345** | 0.740 | 0.257 | 0.082 | 99.8 |
| | | $P_\text{adaptive}$ | 0.376 | 0.709 | 0.286 | 0.118 | 99.6 |
| | | $P_\text{adaptive} + C$ * | 0.348 | **0.744** | **0.241** | **0.069** | **100.0** |
| | 800 | $P_\text{uniform}$ | 0.417 | 0.737 | 0.307 | 0.091 | 99.8 |
| | | $P_\text{adaptive}$ | 0.385 | 0.759 | 0.294 | 0.110 | 99.5 |
| | | $P_\text{adaptive} + C$ * | **0.366** | **0.781** | **0.255** | **0.066** | **100.0** |
| | 1000 | $P_\text{uniform}$ | 0.505 | 0.705 | 0.400 | 0.124 | 98.6 |
| | | $P_\text{adaptive}$ | 0.456 | 0.746 | 0.374 | 0.126 | 98.6 |
| | | $P_\text{adaptive} + C$ * | **0.429** | **0.769** | **0.332** | **0.071** | **99.8** |
| | 1200 | $P_\text{uniform}$ | 0.448 | 0.788 | 0.493 | 0.168 | 95.5 |
| | | $P_\text{adaptive}$ | 0.410 | 0.809 | 0.416 | 0.137 | 98.1 |
| | | $P_\text{adaptive} + C$ * | **0.389** | **0.821** | **0.363** | **0.079** | **99.6** |

**Setup** For the adaptive prior, we set the scale hyperparameters as $(\sigma_{\text{Li}}^{\text{low}}, \sigma_{\text{Li}}^{\text{high}}, \sigma_{\text{frame}}^{\text{low}}, \sigma_{\text{frame}}^{\text{high}}) = (1, 10, 10^{-0.5}, 10^{0.5})$, based on the observation that lithium atoms are generally more diffusive than the frame atoms. For *Corrector* model, we used a maximum noise scale of $\sigma_{\max} = 0.25$ and a small uniform-scale Maxwell–Boltzmann prior with $\sigma = 0.1$. We conducted LiFLOW inference iteratively for $N_{\text{step}} = 25$ steps to simulate dynamics over 25 ps with a time step of $\Delta\tau = 1$ ps. Each inference step involves $N_{\text{flow}} = 10$ flow matching iterations of both *Propagator* and *Corrector* models. During each inference process for a given structure, we terminated when either the maximum number of steps ($N_{\text{step}}$) was reached or the model prediction diverged due to instabilities.

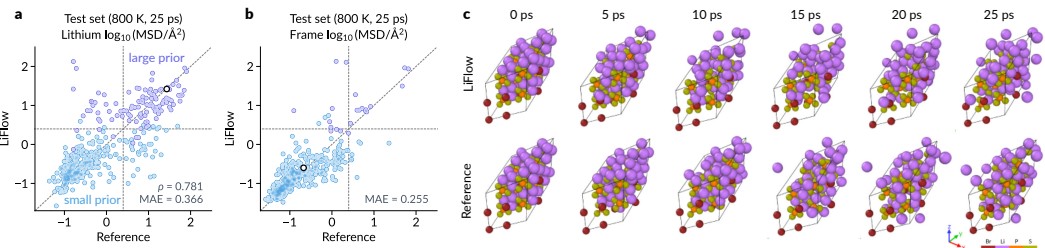

Figure 2: **Universal model inference example. (a, b)** Parity plots comparing the log MSD values for (a) lithium and (b) frame atoms in 800 K, 25 ps simulations across test materials. Data points are colored by their respective prior scales. **(c)** Reference and generated trajectories for $Li_6PS_5Br$ (highlighted points in a and b).

**Reproducing kinetic properties** For LiFLOW baseline model (Table 1, $P_{\text{adaptive}} + C^*$), we consistently observed a Spearman rank correlation of 0.7–0.8 for lithium MSD in unseen test compositions. This indicates the potential of the LiFLOW model for computational screening to identify materials with high lithium diffusivity. The parity plot between log MSD values of reference and LiFLOW-generated trajectories at 800 K, along with visualized example trajectories, is shown in Fig. 2. We observed that the diffusive behavior in well-known solid-state electrolytes, such as $Li_6PS_5Br$ (argyrodite), is accurately reproduced. Note that the stability reported in Table 1 refers to numerical stability—unlike the physical stability of MD trajectories, numerically stable but physically fictitious dynamics can be generated (as reflected in poorer kinetic metrics, see Fig. A4).

### 4.2.2 Ablation Experiments

**Effect of the prior choice** First, in Table 1 (Exp 1), we compare the isotropic prior ($P_{\text{isotropic}}$, $\boldsymbol{\sigma} = \sigma \cdot \mathbf{1}_n$), to the scaled Maxwell–Boltzmann prior ($P_{\text{uniform}}$, $\boldsymbol{\sigma} = \sigma \cdot (k_B T / \boldsymbol{m})^{1/2}$), to evaluate the impact of atom-type-specific scaling on the prior. To focus solely on the relative scale between atoms, we vary the scaling factor $\sigma$ for both the isotropic and Maxwell–Boltzmann priors at a fixed temperature (800 K), then compare the relevant metrics for the optimal $\sigma$ in each case. The best results for isotropic ($\sigma = 10^{-1.5}$) and Maxwell–Boltzmann ($\sigma = 1$) priors are shown in the first row of Table 1, and the results across all scales are provided in Table A3. The scaled Maxwell–Boltzmann prior outperforms the isotropic prior in reproducing all kinetic metrics (log MSD), confirming that the relative scaling of priors among elements is crucial for performance across a wide range of compositions. Additionally, note that the poor performance of direct regression-based displacement prediction (*Regressor*) highlights the necessity of generative modeling.

Next, in Table 1 (Exp 2), we apply the scale for $P_{\text{uniform}}$ determined in the previous experiment to the training and inference on trajectories across all temperatures, and compare to the adaptive scale Maxwell–Boltzmann prior ($P_{\text{adaptive}}$, Eq. (10)). With the exception of the lowest temperature (600 K), where the prior classifier is mostly ineffective (see Fig. A2), the model using the adaptive prior outperforms the one with the uniform scale prior. This suggests that the mixture-of-priors approach effectively guides the flow model in capturing the scale of atomic movements.

**Effect of the *Corrector* model** In Table 1 (Exp 2), we then compare the *Propagator*-only model ($P_{\text{adaptive}}$) and *Propagator + Corrector* model ($P_{\text{adaptive}} + C^*$). We observed improved reproduction of static structural features, indicated by lower RDF MAE, across all temperatures when using the *Corrector* model. Notably, all kinetic metrics also showed improvement with the use of *Corrector*. Since the *Propagator* is a generative model of displacements conditioned on the current time step structure $\boldsymbol{X}_\tau$, correcting errors in the conditional structure improves the accuracy of the predicted cumulative displacements, as reflected in the MSD metric.

## 4.3 AIMD Models

**Setup** Since we are training on the same composition, the *Propagator* error is expected to be smaller than that for the universal dataset, and we accordingly use a smaller maximum noise scale for the *Corrector*. *Corrector* inference can also be simplified by reducing $N_{\text{flow}}$, as detailed in Appendix E.3. For each dataset (LPS and LGPS), a single *Propagator* and *Corrector* model is trained on the first 25 ps of trajectories across temperatures (and polymorph structures for LPS). Additional training and inference settings are provided in Appendix D.4.

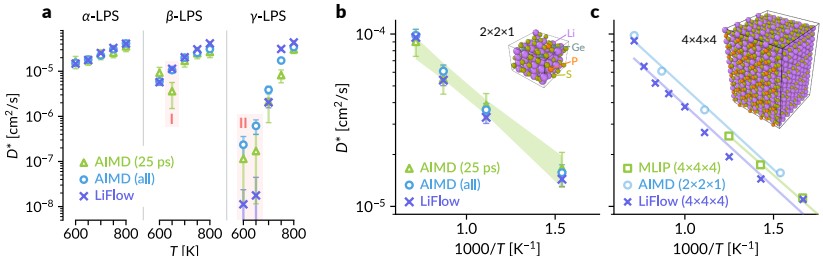

Figure 3: **Reproducing diffusivity from AIMD models.** (a) Lithium $D^*$ for polymorphs of LPS (Li$_3$PS$_4$). Values are derived from AIMD (25 ps training and ∼250 ps full trajectories) and 250 ps LIFLOW inference. (b) Lithium $D^*$ is plotted as a function of $1000/T$ for LGPS (Li$_{10}$GeP$_2$S$_{12}$). $2 \times 2 \times 1$ supercell results from AIMD (25 ps training and ∼150 ps full trajectories) and 150 ps LIFLOW inference. Shaded region represents 95% confidence intervals (CIs) for the Arrhenius fit ($1/T$ vs. $\log D^*(T)$) from 25 ps AIMD data. (c) $4 \times 4 \times 4$ supercell results from fine-tuned MLIP and LIFLOW inference (1 ns).

**Reproducing kinetic properties** Fig. 3a shows the reference diffusivity values for LPS from the AIMD simulations (25 ps for training and ∼250 ps full dataset) alongside the 250 ps LIFLOW inference results. Overall, the LIFLOW results match the order of magnitude of the reference simulations, successfully reproducing the diffusivity differences among the LPS polymorphs. This suggests that the model can detect subtle local structural variations between polymorphs and generate displacements accordingly. In cases where diffusive behavior is expected but not sufficiently captured in a 25 ps trajectory to yield robust diffusivity statistics, LIFLOW can *infill* the correct diffusive dynamics based on other simulations (Fig. 3a, box I). However, when lithium hopping events become exceedingly rare, as in $\gamma$-LPS at lower temperatures, the generative model suffers from mode collapse towards non-diffusive displacements, resulting in an underestimation of $D^*$ (Fig. 3a, box II).

Fig. 3b similarly presents the diffusivity values for LGPS from AIMD simulations and LIFLOW inference. For the $2 \times 2 \times 1$ supercell (Fig. 3b), the temperature dependence of $D^*$, characterized by an activation energy of $E_A = 0.185$ eV, is consistent with the reference AIMD value (0.192 eV, Table 2). Although the 25 ps AIMD $E_A$ (0.173 eV) lies outside the 95% CI of the longer AIMD, LIFLOW success-

Table 2: **Activation energies for LGPS.** Arrhenius fit based on the data in Fig. 3b.

| Method | $E_A$ [eV] | 95% CI [eV] |
|---|---|---|
| AIMD (25 ps) | 0.173 | (0.141, 0.205) |
| AIMD (all) | 0.192 | (0.175, 0.205) |
| LIFLOW | 0.185 | (0.181, 0.190) |

fully matches the longer AIMD result and produces more reliable statistics with lower variance, owing to its extended simulation rollouts.

**Large-scale inference** By modeling the distribution of atomic displacements, the generative model can naturally generalize across different supercell sizes, as indicated by the supercell invariance (Eq. (7)). We evaluated scalability and temperature transferability using a $4 \times 4 \times 4$ supercell, performing LIFLOW inference over 1,000 steps (1 ns), with the resulting $D^*$ values presented in Fig. 3c. For temperatures below the maximum training temperature (1400 K), the LIFLOW model generates stable trajectories that extend far beyond the 25 ps length of the training set trajectories (25 ps). When compared to the reference dynamics from Winter & Gómez-Bombarelli (2023) on LGPS, which used extensive simulations with a fine-tuned MLIP, $D^*$ values predicted by LIFLOW closely match the reference values within the interpolative regime (i.e., the training temperature range). However, as we extend to lower $T$ (higher $1000/T$) beyond the training range, $D^*$ decreases much more slowly than the reference values (Fig. A3), indicating fictitious diffusive behavior when extrapolating to lower $T$. This behavior is expected, as the model was trained primarily on larger displacements of lithium atoms at higher $T$.

**Reproducing structural features** While reproducing kinetics is the main objective of this study, we additionally examined the reproduction of structural features, such as diffusion traces and probability densities of lithium positions. The diffusion traces indicate generalization beyond memorization, with the model exploring symmetrically related sites (Fig. A5). The probability densities are well reproduced, with slight deviations and smoothing at higher temperatures due to the increased complexity of diffusion behavior (Fig. A6). A detailed discussion is provided in Appendix E.2.

### 4.4 COMPUTATIONAL COST

The computation time for 1 ns of inference using the methods investigated in this paper is reported in Table 3. MLIP-based simulations significantly reduce the time required for materials simulations (days to hours), and the LIFLOW model accelerates this even further (hours to seconds). Even taking into account the training time of the LIFLOW model ($\lesssim$ an hour), it remains significantly more efficient than AIMD simulations (see Appendix D.5). Given that AIMD

Table 3: **Prediction speed.** Time required to predict the 1 ns trajectory for LGPS ($^\dagger$extrapolated).

| Method | Supercell | # atoms | Time |
|---|---|---|---|
| AIMD$^\dagger$ | $2 \times 2 \times 1$ | 200 | 340 days |
| MLIP | $2 \times 2 \times 1$ | 200 | 5.8 hrs |
| LIFLOW | $2 \times 2 \times 1$ | 200 | 48 s |
| | $4 \times 4 \times 4$ | 3,200 | 352 s |

scales as $\mathcal{O}(n^3)$ in theory, while both LIFLOW and MLIPs scale as $\mathcal{O}(n)$ for large systems (assuming graphs with radius cutoffs), the LIFLOW model enables efficient large-scale modeling of atomistic dynamics, as demonstrated in this work.

## 5 DISCUSSION

**Conclusion and outlook** We proposed the LIFLOW model, a generative acceleration framework designed to accelerate MD simulations for crystalline materials, with a focus on lithium SSEs. The model consists of two key components: a *Propagator*, which generates atomic displacements for time propagation, and a *Corrector*, which applies denoising. Both components utilize a conditional flow matching scheme, and we introduced a thermally and chemically adaptive prior based on the Maxwell–Boltzmann distribution and modified the PAINN model as a marginal flow approximator, both of which were critical for the accurate reproduction of dynamics. In our analysis of lithium-containing material trajectories, we consistently observed a Spearman rank correlation of 0.7–0.8 for lithium MSD in unseen compositions. This indicates the potential of the LIFLOW model for computational screening to identify materials with high lithium diffusivity. Furthermore, we demonstrated the ability to extend short-length accurate AIMD trajectories by training the LIFLOW model. This allowed us to infill insufficient observations, reproduce accurate temperature dependencies, and maintain high accuracy when scaling up to much larger supercells. Compared to simulations using MLIPs and AIMD, LIFLOW offers significant speedups of 400$\times$ and 600,000$\times$, respectively. This provides a practical means of scaling MD simulations to larger spatiotemporal domains.

**Limitations and future directions** First, although we have demonstrated the importance of designing the prior for the flow matching process, determining the appropriate prior scale remains a hyperparameter. A theoretical analysis of the optimal prior distribution would provide a more principled approach to designing priors tailored to specific acceleration tasks and material systems. This also applies to the choice of time step $\Delta t$: we used a fixed time step based on observation (Appendix A.3), but given the site-to-site hopping nature of atomistic transport, our method may benefit from adaptive or controllable time stepping (e.g., Schreiner et al. (2023)). Additionally, while LIFLOW performs well within the trained temperature range, it struggles to extrapolate beyond the training regime, where system dynamics may differ significantly from the training data. As a result, the current approach lacks the broad generalizability seen in universal MLIP models, which preserve the physical dynamics of systems while approximating the potential energy landscape. To improve reliability and develop a model capable of capturing emergent system behaviors, generative approaches would benefit from incorporating thermodynamic principles more explicitly (Tiwary et al., 2024; Dibak et al., 2022; Herron et al., 2023). Lastly, the accuracy of LIFLOW is inherently limited by the accuracy of the reference dynamics. Given the variety of MD simulation methods and their trade-offs between accuracy and speed, transfer learning or multi-fidelity frameworks could be considered for efficient training in practical applications.

REPRODUCIBILITY STATEMENT

The code and trajectory dataset necessary to reproduce the results will be made publicly available upon acceptance of this work. The sources of the external AIMD trajectories are provided in Appendix C.2.

ETHICS STATEMENT

This work raises ethical considerations related to the general use of machine learning in scientific simulations, particularly in the context of molecular dynamics. While the model presented, LiFlow, is intended to accelerate dynamics simulations for materials science, there is a potential for misuse in harmful applications, such as the development of dangerous materials or chemicals. Although unlikely in the current form of our methodology, we acknowledge the following potential scenarios for misuse, as ML-driven simulations could be misused to design materials with undesirable properties, such as highly reactive compounds that may be hazardous to health or the environment:

- Environmentally harmful materials: Simulations could lead to the creation of materials that, when manufactured or disposed of, could pose long-term environmental risks, such as non-biodegradable or highly polluting compounds.
- Unstable materials: Inaccurate predictions or malicious use of this framework could result in the generation of materials with undesirable or unstable properties, such as those prone to explosive reactions or dangerous degradation.
- Chemical weapons: Simulations may be applied to develop advanced nanomaterials with toxicological risks or harmful capabilities, including those used in biological or chemical warfare.

To mitigate these risks, we commit to working closely with materials experts to ensure responsible usage and oversight of the methodological developments. Additionally, no human subjects, sensitive data, or privacy-related issues are involved in this study, and there are no conflicts of interest or external sponsorships associated with this work.

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

# A    ADDITIONAL BACKGROUND

## A.1    SOLID-STATE ELECTROLYTES

While lithium-ion diffusion in most solid-state materials is generally slow, solid-state electrolytes are a special class of materials in which lithium ions can undergo fast diffusion, often referred to as superionic conductors (Manthiram et al., 2017; Jun et al., 2024a). They serve as a key component for all-solid-state batteries, where enhanced safety is anticipated by replacing flammable organic liquid electrolytes with solid-state alternatives. A major class of solid electrolytes discussed in this work consists of inorganic crystalline materials, which have long-range atomic ordering. In most of these inorganic crystalline solid electrolytes, the anions (Wang et al., 2015) and the non-lithium cations (Jun et al., 2022) remain immobile, selectively permitting the translational motion of the lithium ions. This allows lithium ions to percolate through the crystal structure with a flat energy landscape (or low migration barrier), resulting in a high diffusion coefficient and ionic conductivity.

## A.2    AIMD SIMULATION

***Ab initio* MD**    AIMD simulations are typically carried out using the Born–Oppenheimer molecular dynamics (BOMD) approach (Marx & Hutter, 2009), as defined with the following equation of motion (Eq. (15)) and the time-independent Schrödinger equation (Eq. (16)):

$$\boldsymbol{M}\ddot{\boldsymbol{X}} = -\nabla_{\boldsymbol{X}} \min_{\Psi_0} \langle \Psi_0 | \mathcal{H}(\boldsymbol{X}) | \Psi_0 \rangle, \tag{15}$$

$$E_0 \Psi_0 = \mathcal{H}(\boldsymbol{X}) \Psi_0, \tag{16}$$

where $\mathcal{H}(\boldsymbol{X})$ represents the electronic Hamiltonian, $\Psi_0$ is the ground state wavefunction, and $\langle \Psi_0 | \mathcal{H}(\boldsymbol{X}) | \Psi_0 \rangle$ denotes the quantum mechanical expectation value. At each time step, the ground state wavefunction is obtained by solving Eq. (16), after which the forces are computed as a derivative of the ground state energy with respect to atomic coordinates (Eq. (15)).

For crystalline materials, Eq. (16) is typically solved using Kohn–Sham density functional theory (DFT, Kohn & Sham (1965)), where the ground state energy is expressed as

$$\min_{\Psi_0} \langle \Psi_0 | \mathcal{H}(\boldsymbol{X}) | \Psi_0 \rangle = \min_{\{\phi_i\}} E^{\mathrm{KS}}[\{\phi_i\}; \boldsymbol{X}] \tag{17}$$

with $E^{\mathrm{KS}}$ being the Kohn–Sham energy functional and $\{\phi_i\}$ denoting a set of auxiliary functions called Kohn–Sham orbitals, which depend on the electron positions. The functional optimization in Eq. (17) is equivalent to solving the set of coupled Kohn–Sham equations $H^{\mathrm{KS}} \phi_i = \epsilon_i \phi_i$, where $H^{\mathrm{KS}}$ is the one-particle Hamiltonian and $\{\epsilon_i\}$ are the Kohn–Sham eigenvalues. Solving these eigenvalue equations, which involves diagonalizing the Hamiltonian, scales as $\mathcal{O}(n^3)$ with respect to the system size $n$, defined by the number of atoms (as in the main text) or electrons.

**Thermostat choice**    To maintain a constant temperature $T$ in the system, the dynamics is coupled with an external control mechanism known as a thermostat. We used Nosé–Hoover thermostat (Nosé, 1984; Hoover, 1985) for dataset generation. To avoid thermostat-dependent dynamical artifacts, velocity *scaling* thermostats (e.g., Berendsen, Nosé–Hoover, and stochastic velocity rescaling) should be used instead of velocity *randomization* thermostats (e.g., Langevin and Andersen). The latter may lead to reduced diffusivity values due to rapid decorrelation of velocities (Basconi & Shirts, 2013).

## A.3    TASK DESIGN

**Fixing the volume**    In AIMD simulations for solid electrolytes, or in general when modeling the transport properties of atomistic systems, simulations are typically conducted under the NVT (constant volume) ensemble. Although real materials are often under constant pressure conditions, employing a barostat in simulations to control pressure modifies cell volume, potentially leading to significant changes in particle positions and dynamics (Maginn et al., 2018). In practice, AIMD simulations are initiated after energy minimization of the material structure (with respect to both atomic coordinates and cell dimensions) under the assumption that thermal expansion of the cell does not significantly affect the transport properties.

**Unwrapped coordinates**   In atomistic systems with periodic boundary conditions (PBCs), particles that exit one side of the simulation box effectively reenter from the opposite side. A straightforward way to handle this is to use *wrapped* coordinates, where the positions are continuously confined within the simulation box. However, this introduces jumps in atomic positions during long-range motions, which can distort the calculation of kinetic properties such as MSD and diffusivity. To avoid this, the coordinates must be *unwrapped* before computing such properties. Alternatively, particle positions can be propagated using unwrapped coordinates from the start, without wrapping them back when crossing the cell boundaries.

It is possible to unwrap trajectories during the post-processing of AIMD simulations, assuming that no particles move more than half the cell dimensions between time steps. This condition generally holds for typical AIMD simulations, which use small time steps. However, in the case of LIFLOW modeling in this work, particle displacements can exceed half the box size because (1) we simulate with a much larger time step $\Delta\tau$, and (2) AIMD simulation cells are typically small due to high computational costs (see Appendix A.2). Hence, we use unwrapped coordinates directly when formulating the displacement modeling task for LIFLOW.

**Choice of** $\Delta\tau$   Since the goal of generative displacement modeling in this work is to efficiently accelerate MD simulations, the propagation time step $\Delta\tau$ must be significantly larger than the MD time step $\delta\tau$. However, due to the high cost of generating data, $\Delta\tau$ should not be so large that the modes of atomic displacements are not adequately covered by the training set trajectories.

To determine $\Delta\tau$, we consider the time evolution of lithium MSD for typical lithium-ion solid-state electrolytes. For small $\Delta\tau$ values ($< 0.1$ ps), the MSD grows approximately as MSD $\propto \Delta\tau^{1.42}$, reflecting the ballistic and vibrational motion of lithium ions (He et al., 2018). In this regime, the benefit of generative modeling is limited, as the evolution of atomic positions is closely related to the initial velocities. For larger $\Delta\tau$ ($\gtrsim 1$ ps), the MSD grows linearly as MSD $\propto \Delta\tau$, indicating the onset of diffusive motion, as described by Eq. (13). Given that our training trajectories span 25 ps, we select $\Delta\tau = 1$ ps to ensure that the generative model captures a diverse range of displacement modes present in the training data.

**Units**   The atomic unit system is adopted in this work. Unless stated otherwise, the units are as follows: length is in Å, temperature in K, mass in atomic mass units (u), and energy in eV. For example, the scaling factor for the Maxwell–Boltzmann prior has an implied unit of Å · (eV · K/u)$^{-1/2}$ for converting $(k_BT/\boldsymbol{m})^{1/2}$ into positions.

## B   PROOF FOR PROPOSITION 1

**Proposition 1**   *Given an invariant base distribution $p_0(\boldsymbol{D}_0)$ satisfying Eqs. (3) and (6) and an equivariant conditional vector field $u_t(\boldsymbol{D}_t|\boldsymbol{D}_1)$ with the following properties:*

$$u_t(\boldsymbol{P}\boldsymbol{D}_t|\boldsymbol{P}\boldsymbol{D}_1, \boldsymbol{P}\boldsymbol{X}, \boldsymbol{L}, \boldsymbol{P}\boldsymbol{a}, T) = \boldsymbol{P}u_t(\boldsymbol{D}_t|\boldsymbol{D}_1, \boldsymbol{X}, \boldsymbol{L}, \boldsymbol{a}, T), \qquad \boldsymbol{P} \in S_n \qquad (18)$$

$$u_t(\boldsymbol{D}_t\boldsymbol{R}|\boldsymbol{D}_1\boldsymbol{R}, \boldsymbol{X}\boldsymbol{R}, \boldsymbol{L}\boldsymbol{R}, \boldsymbol{a}, T) = u_t(\boldsymbol{D}_t|\boldsymbol{D}_1, \boldsymbol{X}, \boldsymbol{L}, \boldsymbol{a}, T)\boldsymbol{R}, \qquad \boldsymbol{R} \in O(3) \qquad (19)$$

*the generated conditional probability path $p_{t|1}(\boldsymbol{D}_t|\boldsymbol{D}_1)$ is invariant. Furthermore, given that the data distribution $q(\boldsymbol{D}_1)$ is invariant, the marginal probability path $p_t(\boldsymbol{D}_t)$ is also invariant.*

*Proof.* We will prove for the O(3) symmetry, with a similar approach applying to $S_n$. We omit the conditional variables $(\boldsymbol{X}, \boldsymbol{L}, \boldsymbol{a}, T)$, as their transformations under group actions are implied by those of $\boldsymbol{D}_1$, either remaining invariant or transforming equivariantly. The first part of the proof follows from Theorems 1 and 2 in Köhler et al. (2020), with additional conditional variables. The conditional flow generated by the conditional vector field is

$$\psi_t(\boldsymbol{D}_0|\boldsymbol{D}_1) = \boldsymbol{D}_0 + \int_0^t u_s(\boldsymbol{D}_s|\boldsymbol{D}_1)\,\mathrm{d}s. \qquad (20)$$

Now, we apply $\boldsymbol{R} \in \mathrm{O}(3)$:

$$
\begin{aligned}
\psi_t(\boldsymbol{D}_0\boldsymbol{R}|\boldsymbol{D}_1\boldsymbol{R}) &= \boldsymbol{D}_0\boldsymbol{R} + \int_0^t u_s(\boldsymbol{D}_s\boldsymbol{R}|\boldsymbol{D}_1\boldsymbol{R})\,\mathrm{d}s \\
&= \boldsymbol{D}_0\boldsymbol{R} + \int_0^t u_s(\boldsymbol{D}_s|\boldsymbol{D}_1)\boldsymbol{R}\,\mathrm{d}s \\
&= \left(\boldsymbol{D}_0 + \int_0^t u_s(\boldsymbol{D}_s|\boldsymbol{D}_1)\,\mathrm{d}s\right)\boldsymbol{R} \\
&= \psi_t(\boldsymbol{D}_0|\boldsymbol{D}_1)\boldsymbol{R}.
\end{aligned}
\tag{21}
$$

Thus, the conditional flow $\psi_t$ is also equivariant with respect to $\boldsymbol{R}$. Now, the conditional probability path $p_{t|1}(\boldsymbol{D}_t|\boldsymbol{D}_1)$ is obtained as the pushforward of the prior distribution $p_0$ under $\psi_t$:

$$
p_{t|1}(\boldsymbol{D}_t|\boldsymbol{D}_1) = [\psi_t]_\#p_0(\boldsymbol{D}_0) = p_0\left(\psi_t^{-1}(\boldsymbol{D}_t|\boldsymbol{D}_1)\right)\left|\det \frac{\partial \psi_t^{-1}}{\partial \boldsymbol{D}_t}(\boldsymbol{D}_t|\boldsymbol{D}_1)\right|.
\tag{22}
$$

Again, we apply $\boldsymbol{R} \in \mathrm{O}(3)$:

$$
\begin{aligned}
p_{t|1}(\boldsymbol{D}_t\boldsymbol{R}|\boldsymbol{D}_1\boldsymbol{R}) &= p_0\left(\psi_t^{-1}(\boldsymbol{D}_t\boldsymbol{R}|\boldsymbol{D}_1\boldsymbol{R})\right)\left|\det \frac{\partial \psi_t^{-1}}{\partial (\boldsymbol{D}_t\boldsymbol{R})}(\boldsymbol{D}_t\boldsymbol{R}|\boldsymbol{D}_1\boldsymbol{R})\right| \\
&= p_0\left(\psi_t^{-1}(\boldsymbol{D}_t|\boldsymbol{D}_1)\boldsymbol{R}\right)\left|\det \frac{\partial \psi_t^{-1}}{\partial (\boldsymbol{D}_t\boldsymbol{R})}(\boldsymbol{D}_t\boldsymbol{R}|\boldsymbol{D}_1\boldsymbol{R})\right| \\
&= p_0\left(\psi_t^{-1}(\boldsymbol{D}_t|\boldsymbol{D}_1)\right)|\det \boldsymbol{I}_n \otimes \boldsymbol{R}|\left|\det \frac{\partial \psi_t^{-1}}{\partial \boldsymbol{D}_t}(\boldsymbol{D}_t|\boldsymbol{D}_1)\right||\det \boldsymbol{I}_n \otimes \boldsymbol{R}|^{-1} \\
&= p_0\left(\psi_t^{-1}(\boldsymbol{D}_t|\boldsymbol{D}_1)\right)\left|\det \frac{\partial \psi_t^{-1}}{\partial \boldsymbol{D}_t}(\boldsymbol{D}_t|\boldsymbol{D}_1)\right| \\
&= p_{t|1}(\boldsymbol{D}_t|\boldsymbol{D}_1),
\end{aligned}
\tag{23}
$$

where we used the fact that $|\det \boldsymbol{I}_n \otimes \boldsymbol{R}| = |\det \boldsymbol{R}|^n = 1$. Therefore, the resulting conditional probability path $p_{t|1}$ is also invariant with respect to $\boldsymbol{R}$.

Now, for the marginal probability $p_t(\boldsymbol{D}_t) = \int p_{t|1}(\boldsymbol{D}_t|\boldsymbol{D}_1)q(\boldsymbol{D}_1)\mathrm{d}\boldsymbol{D}_1$,

$$
\begin{aligned}
p_t(\boldsymbol{D}_t\boldsymbol{R}) &= \int p_{t|1}(\boldsymbol{D}_t\boldsymbol{R}|\boldsymbol{D}_1\boldsymbol{R})q(\boldsymbol{D}_1\boldsymbol{R})\,\mathrm{d}(\boldsymbol{D}_1\boldsymbol{R}) \\
&= \int p_{t|1}(\boldsymbol{D}_t|\boldsymbol{D}_1)q(\boldsymbol{D}_1)\,|\det \boldsymbol{I}_n \otimes \boldsymbol{R}|\,\mathrm{d}\boldsymbol{D}_1 \\
&= \int p_{t|1}(\boldsymbol{D}_t|\boldsymbol{D}_1)q(\boldsymbol{D}_1)\,\mathrm{d}\boldsymbol{D}_1 \\
&= p_t(\boldsymbol{D}_t),
\end{aligned}
\tag{24}
$$

which concludes the proof of the invariance of the marginal $p_t$. $\qquad\square$

## C  DATASET DETAILS

### C.1  UNIVERSAL DATASET

We fetched 4,186 lithium-containing structures from Materials Project (Jain et al., 2013) with the criteria of (1) more than 10% of the atoms are lithium, (2) band gap $> 2$ eV, and (3) energy over the convex hull $< 0.1$ eV/atom. These criteria are designed to sample various modes of lithium-ion dynamics across different compositions, while maintaining minimal requirements for the solid-state electrolytes. After building a supercell of the structure in order to ensure that each dimension is larger than 9 Å and minimizing the structure, we conducted NVT MD simulations with MACE-MP-0 small model (Batatia et al., 2024) at 600, 800, 1000, and 1200 K for each structure. The initial velocities were assigned according to the temperature, and the system was propagated for 25 ps with the time step of 1 fs (25,000 steps) using Nosé–Hoover dynamics (Nosé, 1984; Hoover, 1985) as implemented in ASE (Larsen et al., 2017). We recorded the atom positions every ten steps.

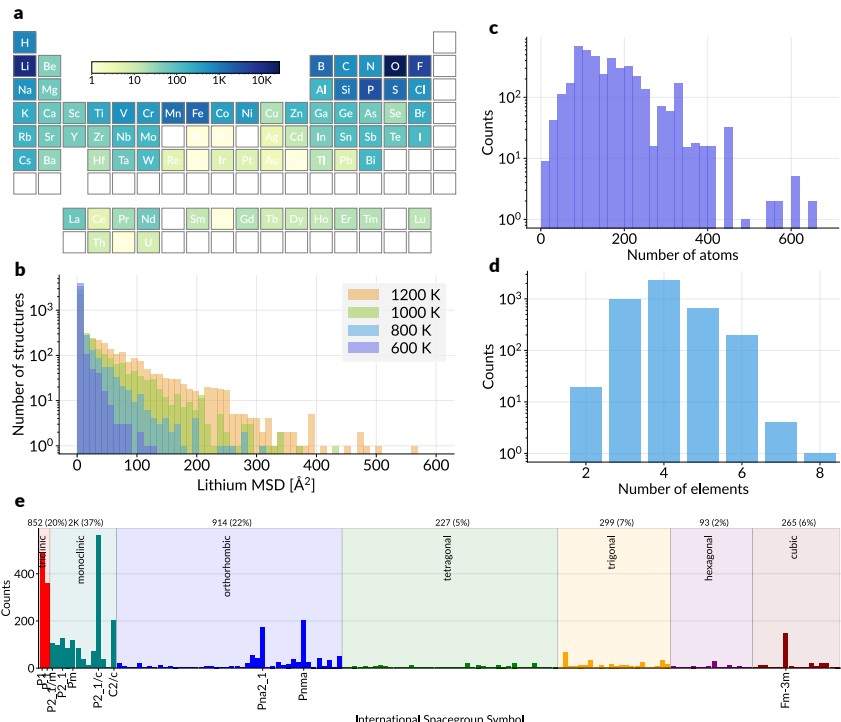

Figure A1: **Dataset statistics.** **(a)** Elemental count distribution across the unit cells of the structures in the dataset. **(b)** Histogram of lithium MSD values from 25-ps MD simulations at different temperatures. **(c)** Distribution of atom counts (in the constructed supercell) per structure. **(d)** Distribution of element counts per structure. **(e)** Space group distribution of the structures (visualized with Pymatviz (Riebesell et al., 2022)).

### C.2 AIMD DATASETS

**LPS dataset** Among the three LPS polymorphs, $\alpha$- and $\beta$-LPS are fast lithium-ion conductors that remain stable at high temperatures, whereas the $\gamma$-phase is a slower lithium-ion conductor (Kimura et al., 2023; Lee et al., 2023). These polymorphs provide an excellent system to evaluate the capability of our model, as their crystal structures are quite similar—primarily differentiated by the orientation of the $PS_4$ tetrahedra and the corresponding lithium-ion sites—yet they exhibit drastically different lithium transport properties. We obtained the LPS trajectories from Jun et al. (2024b) directly from the authors. Supercell sizes of $2 \times 2 \times 2$, $1 \times 2 \times 2$, and $2 \times 2 \times 2$ were used for $\alpha$-, $\beta$-, and $\gamma$-$Li_3PS_4$, respectively. For each structure, five trajectories at temperatures of 600, 650, 700, 750, and 800 K were used. The reference trajectories used a time step of $\delta\tau = 2$ fs, which we subsampled every five steps to reduce redundancy in the training and test datasets. We set the LIFLOW time step $\Delta\tau$ to 500 steps (1 ps).

**LGPS dataset** LGPS is a prototypical lithium superionic conductor discovered in 2011 (Kamaya et al., 2011). We utilized AIMD trajectories for LGPS from López et al. (2024), accessible at https://superionic.upc.edu/. The reference simulations employed a time step of $\delta\tau = 1.5$ fs, with snapshots recorded every ten steps (15 fs). To align with this, we set the LIFLOW time step $\Delta\tau$ to 670 steps (1.005 ps).

## D MODEL AND TRAINING

### D.1 PRIOR SELECTOR

The prior selector model $\sigma_{\mathcal{S}}(\mathcal{M}_0, T)$ for species $\mathcal{S}$ (lithium or frame) is a binary classifier that predicts whether the atom of the given species $\mathcal{S}$ will exhibit large or small displacements based on the initial structure of materials. The same training and test splits were used for the universal dataset.

Labels for large and small displacements were determined by the criterion $\text{MSD}_\mathcal{S}/\tau < 0.1$ Å$^2$/ps, computed over the reference simulation ($\tau = 25$ ps). The input features for the classifier are the atomic invariant features (128 dimensions) averaged over atoms of $\mathcal{S}$, extracted from a pre-trained MACE-MP-0 small model (Batatia et al., 2024) given the initial structure $(\boldsymbol{X}_0, \boldsymbol{L}, \boldsymbol{a})$, along with the temperature ($T/1000$ K, a scalar). These features are concatenated and fed into a multi-layer perceptron with hidden layers of size 32 and 16, which is trained on the training set materials. The histograms of the $\text{MSD}_\mathcal{S}/\tau$ distribution annotated with predicted labels are reported in Fig. A2.

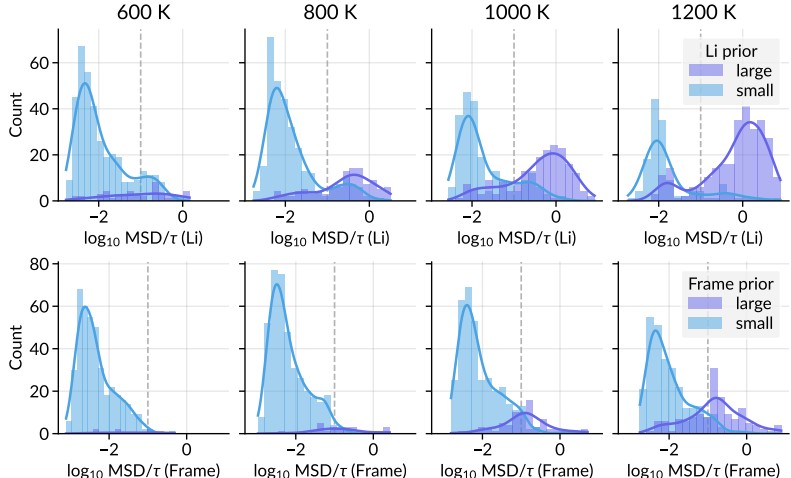

Figure A2: **Prior selector model performance.** Histogram of the target values, $\log_{10}(\text{MSD}_\mathcal{S}/\tau)$, for lithium and frame atoms, colored by the predicted prior scale (small or large) for the test set materials. The intended classification threshold $(-1.0)$ is marked by a vertical dotted line.

## D.2 FLOW MODEL ARCHITECTURE

We adapt the PAINN model (Schütt et al., 2021) to parametrize the marginal flow approximator $v_\theta(\boldsymbol{D}_t|\boldsymbol{X}_\tau, \boldsymbol{L}, \boldsymbol{a}, T)$ for both the *Propagator* and *Corrector*. Schreiner et al. (2023) employed a modified version of the PAINN model, named CHIROPAINN, for a similar task for small biomolecules, introducing cross products during message passing in order to break reflection symmetry. Their modification was necessary due to their use of coarse-grained protein representation (C$_\alpha$ coordinates), where the mirror image of a C$_\alpha$ trace does not correspond to the mirror image of the full-atom structure. In contrast, we represent the material structure using all atomic coordinates without coarse-graining, preserving the reflection symmetry of the atomistic system. As a result, we chose to modify the original PAINN architecture instead of CHIROPAINN.

**Node input features** The model employs a learnable atomic embedding function, $f_{\text{atom}} : \mathcal{A} \to \mathbb{R}^{d_f}$, to map atomic species to feature vectors, where $d_f$ is the feature dimension. For continuous values, the embedding function $f_{\text{cont}} : \mathbb{R} \to \mathbb{R}^{d_f/2}$ is defined using a sinusoidal encoding:

$$[f_{\text{cont}}(x)]_i = \begin{cases} \sin\left(2\pi f_{\lfloor i/2 \rfloor} x\right) & i \text{ odd}, \\ \cos\left(2\pi f_{\lfloor i/2 \rfloor} x\right) & i \text{ even}, \end{cases} \tag{25}$$

where $f_i$ ($i \in [\![1, d_f/4]\!]$) are frequencies sampled from a standard normal distribution $\mathcal{N}(0, 1^2)$ and fixed during training. The invariant node embedding for atom $j$ is computed as $f_{\text{atom}}(a_j) + (f_{\text{cont}}(T/1000) \oplus f_{\text{cont}}(t))$, for temperature $T$ and flow matching time $t$. Rather than initializing equivariant node features to zeros as in the original model, they are initialized from the current step displacement as $\boldsymbol{D}_t \otimes \boldsymbol{w} \in \mathbb{R}^{n \times 3 \times d_f}$, where $\boldsymbol{w} \in \mathbb{R}^{d_f}$ is a learnable weight vector.

**Message passing** For clarity and ease of comparison, we use the notation from the PAINN paper (Schütt et al., 2021) for this part. As described in the main text, we leverage information from two sets of coordinates, $\boldsymbol{X}_\tau$ and $\boldsymbol{X}_\tau + \boldsymbol{D}_t$, during message passing. A similar approach using two

sets of edge information was previously employed by Hsu et al. (2024). To simplify computation, we define the edges using a radius cutoff graph based on $\boldsymbol{X}_\tau$, avoiding the need to reconstruct the neighbor graph at each flow matching step $t$. When expanding distances into radial basis functions, we shift the distance by 0.5 Å. Unlike physically realistic atomistic systems, during flow integration, the structure $\boldsymbol{X}_\tau + \boldsymbol{D}_t$ may experience atomic clashes. Since Bessel function values change most significantly at small radii, shifting the distances helps reduce variance in the edge features.

In the message functions that use continuous-filter convolutions, we apply elementwise addition of the filters corresponding to the two distances, $\|\vec{r}_{ij,1}\|$ and $\|\vec{r}_{ij,2}\|$. To avoid introducing unintended permutation symmetry between two geometries, we use two distinct filters ($\mathcal{W}'_{vs,k}$ in Eq. (27b)) for the respective unit vector directions. The invariant message update Eq. (26a) (Eq. (7) in the original paper) is modified as Eq. (26b):

$$\Delta \mathbf{s}_i^m = \sum_j \phi_s(\mathbf{s}_j) \circ \mathcal{W}_s(\|\vec{r}_{ij}\|), \tag{26a}$$

$$\Delta \mathbf{s}_i^m = \sum_j \phi_s(\mathbf{s}_j) \circ \left[\mathcal{W}_s(\|\vec{r}_{ij,1}\|) + \mathcal{W}_s(\|\vec{r}_{ij,2}\|)\right], \tag{26b}$$

and the equivariant message update Eq. (27a) (Eq. (8)) in the original paper) is modified as Eq. (27b):

$$\Delta \vec{\mathbf{v}}_i^m = \sum_j \vec{\mathbf{v}}_j \circ \phi_{vv}(\mathbf{s}_j) \circ \mathcal{W}_{vv}(\|\vec{r}_{ij}\|) + \sum_j \phi_{vs}(\mathbf{s}_j) \circ \mathcal{W}'_{vs}(\|\vec{r}_{ij}\|) \frac{\vec{r}_{ij}}{\|\vec{r}_{ij}\|}, \tag{27a}$$

$$\Delta \vec{\mathbf{v}}_i^m = \sum_j \vec{\mathbf{v}}_j \circ \phi_{vv}(\mathbf{s}_j) \circ \left[\mathcal{W}_{vv}(\|\vec{r}_{ij,1}\|) + \mathcal{W}_{vv}(\|\vec{r}_{ij,2}\|)\right]$$

$$+ \sum_{k \in \{1,2\}} \sum_j \phi_{vs,k}(\mathbf{s}_j) \circ \left[\mathcal{W}'_{vs,k}(\|\vec{r}_{ij,1}\|) + \mathcal{W}'_{vs,k}(\|\vec{r}_{ij,2}\|)\right] \frac{\vec{r}_{ij,k}}{\|\vec{r}_{ij,k}\|}. \tag{27b}$$

Table A1: **Effect of PAINN modification.** Evaluation metrics for the *Propagator* model using a uniform scale ($\sigma = 1$) Maxwell–Boltzmann prior distribution. Standard deviations are from three independent generations.

| Train $T$ (K) | Inference $T$ (K) | Model | log MSD$_{\text{Li}}$ MAE ($\downarrow$) | log MSD$_{\text{Li}}$ $\rho$ ($\uparrow$) | log MSD$_{\text{frame}}$ MAE ($\downarrow$) | Stable traj. % ($\uparrow$) |
|---|---|---|---|---|---|---|
| 800 | 800 | PAINN | $0.976_{\pm 0.008}$ | $0.344_{\pm 0.005}$ | $1.217_{\pm 0.009}$ | $38.1_{\pm 1.3}$ |
| | | Modified PAINN | $\mathbf{0.396}_{\pm 0.006}$ | $\mathbf{0.779}_{\pm 0.009}$ | $\mathbf{0.274}_{\pm 0.003}$ | $\mathbf{99.4}_{\pm 0.1}$ |

**Performance comparison**  Since we use $\boldsymbol{D}_t$ to initialize the vector node features, the additional positional input $\boldsymbol{X}_\tau + \boldsymbol{D}_t$ could be omitted without losing information, allowing the use of the original PAINN model. Table A1 presents a comparison of the metrics from Table 1 between the original and modified PAINN models. The results show a significant difference between the two models, highlighting the importance of incorporating the intermediate structure $\boldsymbol{X}_\tau + \boldsymbol{D}_t$.

### D.3  TRAINING ALGORITHMS

The training algorithms for the *Propagator* and *Corrector* are shown in Algorithms A1 and A2, respectively. When training on the universal dataset, material compositions are sampled uniformly by assigning a sampling weight inversely proportional to the number of materials in the training set with that specific composition.

### D.4  TRAINING AND INFERENCE HYPERPARAMETERS

The training and model hyperparameters are summarized in Table A2. Additionally, validation loss was evaluated every 1,250 training steps, with early stopping triggered if the validation loss did not improve after ten evaluations. The model parameters corresponding to the lowest validation loss were used for inference.

---

**Algorithm A1:** LɪFʟᴏᴡ *Propagator* Training

---

**Input:** Dataset of time-separated material structures $\mathcal{D}$
**Output:** Optimized *Propagator* parameter $\theta$

**while** *Training* **do**
    Sample data $(\boldsymbol{X}_\tau, \boldsymbol{X}_{\tau+\Delta\tau}, \boldsymbol{L}, \boldsymbol{a}, T) \sim \mathcal{D}$
    Sample flow time $t \sim \mathcal{U}(t; 0, 1)$
    Sample *Propagator* prior $\boldsymbol{D}_0 \sim \mathcal{N}(\boldsymbol{D}_0; \boldsymbol{0}, \mathrm{diag}(\boldsymbol{\sigma})^2 \otimes \boldsymbol{I}_3)$
    $\boldsymbol{D}_1 \leftarrow \boldsymbol{X}_{\tau+\Delta\tau}$          // True displacements
    $\boldsymbol{D}_t \leftarrow (1-t)\boldsymbol{D}_0 + t\boldsymbol{D}_1$    // Interpolated displacements (Eq. (11))
    $u_t(\boldsymbol{D}_t|\boldsymbol{D}_1) \leftarrow (\boldsymbol{D}_1 - \boldsymbol{D}_t)/(1-t)$    // Conditional flow (Eq. (11))
    $v_t(\boldsymbol{D}_t; \theta) \leftarrow Propagator(\boldsymbol{D}_t, \boldsymbol{X}_\tau, \boldsymbol{L}, \boldsymbol{a}, T, t; \theta)$
    $\mathcal{L}_{\mathrm{CFM}}(\theta) \leftarrow \|v_t(\boldsymbol{D}_t; \theta) - u_t(\boldsymbol{D}_t|\boldsymbol{D}_1)\|^2$    // CFM regression objective (Eq. (2))
    $\theta \leftarrow \mathrm{Update}(\theta, \nabla_\theta \mathcal{L}_{\mathrm{CFM}}(\theta))$    // Parameter update

---

**Algorithm A2:** LɪFʟᴏᴡ *Corrector* Training

---

**Input:** Dataset of time-separated material structures $\mathcal{D}$
**Output:** Optimized *Corrector* parameter $\theta$

**while** *Training* **do**
    Sample data $(\cdot, \boldsymbol{X}_\tau, \boldsymbol{L}, \boldsymbol{a}, T) \sim \mathcal{D}$
    Sample flow time $t \sim \mathcal{U}(t; 0, 1)$
    Sample *Corrector* prior $\boldsymbol{D}_0 \sim \mathcal{N}(\boldsymbol{D}_0; \boldsymbol{0}, \mathrm{diag}(\boldsymbol{\sigma})^2 \otimes \boldsymbol{I}_3)$
    Sample noise scale $\boldsymbol{\sigma}' \sim \mathcal{U}(\boldsymbol{\sigma}'; \boldsymbol{0}, \sigma_{\max} \boldsymbol{1}_n)$
    Sample positional noise displacement $\boldsymbol{D}|\boldsymbol{\sigma}' \sim \mathcal{N}(\boldsymbol{D}; \boldsymbol{0}, \mathrm{diag}(\boldsymbol{\sigma}')^2 \otimes \boldsymbol{I}_3)$
    $\tilde{\boldsymbol{X}}_\tau \leftarrow \boldsymbol{X}_\tau + \boldsymbol{D}$        // Noisy positions
    $\boldsymbol{D}_1 \leftarrow -\boldsymbol{D}$        // True denoising displacements
    $\boldsymbol{D}_t \leftarrow (1-t)\boldsymbol{D}_0 + t\boldsymbol{D}_1$    // Interpolated displacements (Eq. (11))
    $u_t(\boldsymbol{D}_t|\boldsymbol{D}_1) \leftarrow (\boldsymbol{D}_1 - \boldsymbol{D}_t)/(1-t)$    // Conditional flow (Eq. (11))
    $v_t(\boldsymbol{D}_t; \theta) \leftarrow Corrector(\boldsymbol{D}_t, \tilde{\boldsymbol{X}}_\tau, \boldsymbol{L}, \boldsymbol{a}, T, t; \theta)$
    $\mathcal{L}_{\mathrm{CFM}}(\theta) \leftarrow \|v_t(\boldsymbol{D}_t; \theta) - u_t(\boldsymbol{D}_t|\boldsymbol{D}_1)\|^2$    // CFM regression objective (Eq. (2))
    $\theta \leftarrow \mathrm{Update}(\theta, \nabla_\theta \mathcal{L}_{\mathrm{CFM}}(\theta))$    // Parameter update

---

For *Propagator* in the AIMD dataset, we replaced the prior classifier with fixed prior scale parameters for each temperature, determined based on the MSD values from the training trajectories. Additionally, in both LGPS and LPS, the frame atoms did not exhibit diffusive behavior, so we applied a uniform prior scale for these atoms. The prior scales used were $(\sigma_{\mathrm{Li}}^{\mathrm{small}}, \sigma_{\mathrm{Li}}^{\mathrm{large}}) = (1, 10)$ for lithium atoms, and $\sigma_{\mathrm{frame}} = 0.5$ for LGPS and $1$ for LPS. For the *Corrector*, we set the maximum noise scale to $\sigma_{\max} = 0.1$ for the $2 \times 2 \times 1$ supercell of LGPS and for all LPS experiments. For the larger $4 \times 4 \times 4$ supercell inference in LGPS, we used a *Corrector* trained with $\sigma_{\max} = 0.2$ to improve trajectory stability.

We performed LɪFʟᴏᴡ inference for $N_{\mathrm{step}} = 150$ steps in the $2 \times 2 \times 1$ LGPS simulations and $N_{\mathrm{step}} = 1000$ steps in the $4 \times 4 \times 4$ LGPS simulations, with a time step of $\Delta\tau = 1.005$ ps. This corresponds to total simulation times of 150.75 ps and 1.005 ns, respectively. For the LPS simulations, we used $N_{\mathrm{step}} = 250$ steps with a time step of $\Delta\tau = 1$ ps, resulting in a total simulation time of 250 ps. We used Euler integration with $N_{\mathrm{flow}} = 10$ steps for all experiments. For AIMD simulations, since the *Propagator* error is relatively small, *Corrector* inference can be simplified without impacting simulation results—for example, by reducing $N_{\mathrm{flow}}$ to 1. Details of these ablation studies are provided in Appendix E.3.

### D.5 Implementation Details and Computational Cost

We implemented the LɪFʟᴏᴡ model using PyTorch (Paszke et al., 2019) and PyG (Fey & Lenssen, 2019) libraries. For MLIP-based simulations, we utilized MACE-MP-0 (mace-torch package,

Table A2: Hyperparameters for training the *Propagator* and *Corrector* models.

| Parameter | Value |
|---|---|
| Feature dimension | 64 |
| Radial basis functions | 20 |
| Message passing layers | 3 |
| Cutoff distance | 5.0 |
| Offset distance | 0.5 |
| Optimizer | Adam (Kingma & Ba, 2014) |
| Learning rate | 0.0003 |
| Gradient clipping norm | 10.0 |
| Batch size | 16 |
| Maximum training steps | 125,000 |

Batatia et al. (2024)) in combination with ASE (Larsen et al., 2017). Bayesian analysis of diffusivity and activation energy was performed using the `kinisi` package (McCluskey et al., 2024b). Training and inference of LiFlow models were performed using a single NVIDIA RTX A5000 GPU. The training process for the *Propagator* and *Corrector* models, using early stopping, typically lasts between 45,000 and 70,000 steps. This corresponds to approximately 40–60 minutes of training, extending to up to two hours if the maximum step budget is reached. For AIMD simulation in Table 3, we used the $\Gamma$-point only version of VASP (`vasp_gam`, Hafner (2008)) with 48 cores of an Intel Xeon Gold 8260 CPU. The same input files used in the LGPS AIMD simulations were utilized for the benchmark.

# E    ADDITIONAL RESULTS

## E.1    UNIVERSAL MODEL: EXTENDED RESULTS AND CASE STUDIES

The comparison between the isotropic prior and the scaled Maxwell–Boltzmann priors are shown in Table A3 (Page 24). The standard deviations for results in Table 1 are reported in Table A4 (Page 25). Examples of LiFlow model inference trajectories for the universal model are presented in Fig. A4 (Page 26).

Table A3: **Effect of prior design.** Comparison between the isotropic prior and the scaled Maxwell–Boltzmann prior using different scale multipliers. Only the propagator model was used, and was trained and tested on 800 K trajectories.

| Prior | Scale multiplier ($\sigma$) | $\log \text{MSD}_{\text{Li}}$ MAE ($\downarrow$) | $\log \text{MSD}_{\text{Li}}$ $\rho$ ($\uparrow$) | $\log \text{MSD}_{\text{frame}}$ MAE ($\downarrow$) | Stable traj. % ($\uparrow$) |
|---|---|---|---|---|---|
| Isotropic | $10^{-2}$ | $0.726_{\pm 0.012}$ | $0.550_{\pm 0.008}$ | $0.900_{\pm 0.007}$ | $99.9_{\pm 0.1}$ |
| | $10^{-1.5}$ | $\underline{0.498}_{\pm 0.003}$ | $\underline{0.753}_{\pm 0.008}$ | $\underline{0.318}_{\pm 0.008}$ | $98.6_{\pm 0.2}$ |
| | $10^{-1}$ | $0.531_{\pm 0.008}$ | $0.713_{\pm 0.008}$ | $0.454_{\pm 0.012}$ | $95.9_{\pm 0.2}$ |
| | $10^{-0.5}$ | $0.551_{\pm 0.009}$ | $0.723_{\pm 0.004}$ | $0.470_{\pm 0.001}$ | $\underline{\textbf{100.0}}_{\pm 0.0}$ |
| | $10^{0}$ | $0.626_{\pm 0.005}$ | $0.712_{\pm 0.004}$ | $0.408_{\pm 0.002}$ | $\underline{\textbf{100.0}}_{\pm 0.0}$ |
| Maxwell–Boltzmann | $10^{-1}$ | $0.694_{\pm 0.002}$ | $0.563_{\pm 0.007}$ | $0.653_{\pm 0.003}$ | $88.1_{\pm 1.5}$ |
| | $10^{-0.5}$ | $0.511_{\pm 0.004}$ | $0.682_{\pm 0.006}$ | $0.419_{\pm 0.004}$ | $99.8_{\pm 0.2}$ |
| | $10^{0}$ | $\textbf{0.396}_{\pm 0.006}$ | $\textbf{0.779}_{\pm 0.009}$ | $\textbf{0.274}_{\pm 0.003}$ | $99.4_{\pm 0.1}$ |
| | $10^{0.5}$ | $0.654_{\pm 0.002}$ | $0.694_{\pm 0.006}$ | $0.447_{\pm 0.005}$ | $99.4_{\pm 0.1}$ |
| | $10^{1}$ | $0.577_{\pm 0.007}$ | $0.709_{\pm 0.009}$ | $0.339_{\pm 0.007}$ | $\underline{99.9}_{\pm 0.1}$ |

## E.2    LGPS AIMD MODEL: TEMPERATURE EXTRAPOLATION AND REPRODUCING STRUCTURAL FEATURES

The temperature extrapolation results, which extend from Fig. 3c, are displayed in Fig. A3. As we extend to lower $T$ (higher $1000/T$) beyond the training range, $D^*$ decreases much more slowly than

Table A4: **Results for the universal model.** Evaluation metrics for different *Propagator* priors (isotropic and uniform/adaptive scale Maxwell–Boltzmann) with or without the *Corrector*. *Regressor*[†]: non-generative, directly predicting displacements. Standard deviations are from three independent generations.

| Train $T$ (K) | Inference $T$ (K) | Model | log MSD$_{Li}$ MAE ($\downarrow$) | log MSD$_{Li}$ $\rho$ ($\uparrow$) | log MSD$_{frame}$ MAE ($\downarrow$) | RDF MAE ($\downarrow$) | Stable traj. % ($\uparrow$) |
|---|---|---|---|---|---|---|---|
| 800 | 800 | *Regressor*[†] | 1.636 | 0.535 | 0.876 | 0.416 | 90.2 |
|  |  | $P_{\text{isotropic}}$ | $0.498_{\pm 0.003}$ | $0.753_{\pm 0.008}$ | $0.318_{\pm 0.008}$ | $0.113_{\pm 0.0020}$ | $98.6_{\pm 0.2}$ |
|  |  | $P_{\text{uniform}}$ | $\mathbf{0.396}_{\pm 0.006}$ | $\mathbf{0.779}_{\pm 0.009}$ | $\mathbf{0.274}_{\pm 0.003}$ | $\mathbf{0.084}_{\pm 0.0004}$ | $\mathbf{99.4}_{\pm 0.1}$ |
| All | 600 | $P_{\text{uniform}}$ | $\mathbf{0.345}_{\pm 0.003}$ | $0.740_{\pm 0.009}$ | $0.257_{\pm 0.006}$ | $0.082_{\pm 0.0001}$ | $99.8_{\pm 0.2}$ |
|  |  | $P_{\text{adaptive}}$ | $0.376_{\pm 0.005}$ | $0.709_{\pm 0.003}$ | $0.286_{\pm 0.001}$ | $0.118_{\pm 0.0002}$ | $99.6_{\pm 0.2}$ |
|  |  | $P_{\text{adaptive}} + C$ | $0.348_{\pm 0.004}$ | $\mathbf{0.744}_{\pm 0.012}$ | $\mathbf{0.241}_{\pm 0.002}$ | $\mathbf{0.069}_{\pm 0.0001}$ | $\mathbf{100.0}_{\pm 0.0}$ |
|  | 800 | $P_{\text{uniform}}$ | $0.417_{\pm 0.007}$ | $0.737_{\pm 0.011}$ | $0.307_{\pm 0.003}$ | $0.091_{\pm 0.0005}$ | $99.8_{\pm 0.2}$ |
|  |  | $P_{\text{adaptive}}$ | $0.385_{\pm 0.004}$ | $0.759_{\pm 0.008}$ | $0.294_{\pm 0.001}$ | $0.110_{\pm 0.0004}$ | $99.5_{\pm 0.0}$ |
|  |  | $P_{\text{adaptive}} + C$ | $\mathbf{0.366}_{\pm 0.005}$ | $\mathbf{0.781}_{\pm 0.005}$ | $\mathbf{0.255}_{\pm 0.004}$ | $\mathbf{0.066}_{\pm 0.0000}$ | $\mathbf{100.0}_{\pm 0.0}$ |
|  | 1000 | $P_{\text{uniform}}$ | $0.505_{\pm 0.011}$ | $0.705_{\pm 0.008}$ | $0.400_{\pm 0.007}$ | $0.124_{\pm 0.0006}$ | $98.6_{\pm 0.2}$ |
|  |  | $P_{\text{adaptive}}$ | $0.456_{\pm 0.024}$ | $0.746_{\pm 0.008}$ | $0.374_{\pm 0.003}$ | $0.126_{\pm 0.0004}$ | $98.6_{\pm 0.6}$ |
|  |  | $P_{\text{adaptive}} + C$ | $\mathbf{0.429}_{\pm 0.003}$ | $\mathbf{0.769}_{\pm 0.006}$ | $\mathbf{0.332}_{\pm 0.002}$ | $\mathbf{0.071}_{\pm 0.0001}$ | $\mathbf{99.8}_{\pm 0.1}$ |
|  | 1200 | $P_{\text{uniform}}$ | $0.448_{\pm 0.006}$ | $0.788_{\pm 0.003}$ | $0.493_{\pm 0.003}$ | $0.168_{\pm 0.0013}$ | $95.5_{\pm 0.5}$ |
|  |  | $P_{\text{adaptive}}$ | $0.410_{\pm 0.002}$ | $0.809_{\pm 0.003}$ | $0.416_{\pm 0.003}$ | $0.137_{\pm 0.0004}$ | $98.1_{\pm 0.6}$ |
|  |  | $P_{\text{adaptive}} + C$ | $\mathbf{0.389}_{\pm 0.005}$ | $\mathbf{0.821}_{\pm 0.004}$ | $\mathbf{0.363}_{\pm 0.003}$ | $\mathbf{0.079}_{\pm 0.0002}$ | $\mathbf{99.6}_{\pm 0.1}$ |

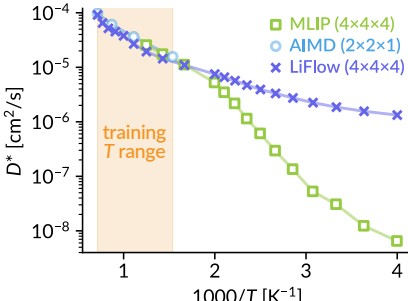

Figure A3: **Temperature extrapolation.** Lithium $D^*$ is plotted as a function of $1000/T$ for LGPS ($\text{Li}_{10}\text{GeP}_2\text{S}_{12}$), extending the data from Fig. 3c to lower temperatures (higher $1000/T$).

the reference values, indicating fictitious diffusive behavior when extrapolating to lower $T$. This behavior is expected, as the model was trained primarily on larger displacements of lithium atoms at higher $T$.

The diffusion trace in Fig. A5 (Page 27) shows that the generated dynamics and the reference dynamics explore different but symmetrically related sites in unwrapped coordinates. This confirms that the model is not merely memorizing the reference dynamics but is generalizing to physically equivalent configurations. Additionally, 2-D log probability densities[2] of lithium atoms are plotted along the $x$–$y$ and $y$–$z$ planes in Fig. A6 (Page 27). The log densities are accurately reproduced at lower temperatures, but deviate at higher temperatures, becoming noisier for LiFLOW, which results in a smoothing of the (free) energy landscape. As the displacements due to diffusion become larger and more varied at higher temperatures, we expect it to be more challenging to achieve high accuracy for static structural features under these conditions.

---

[2]The negative log density, scaled by $k_B T$, $F(\boldsymbol{x}) = -k_B T \log p(\boldsymbol{x})$, is also known as the *potential of mean force* (PMF) or the *free energy surface* in the chemistry and physics literature.

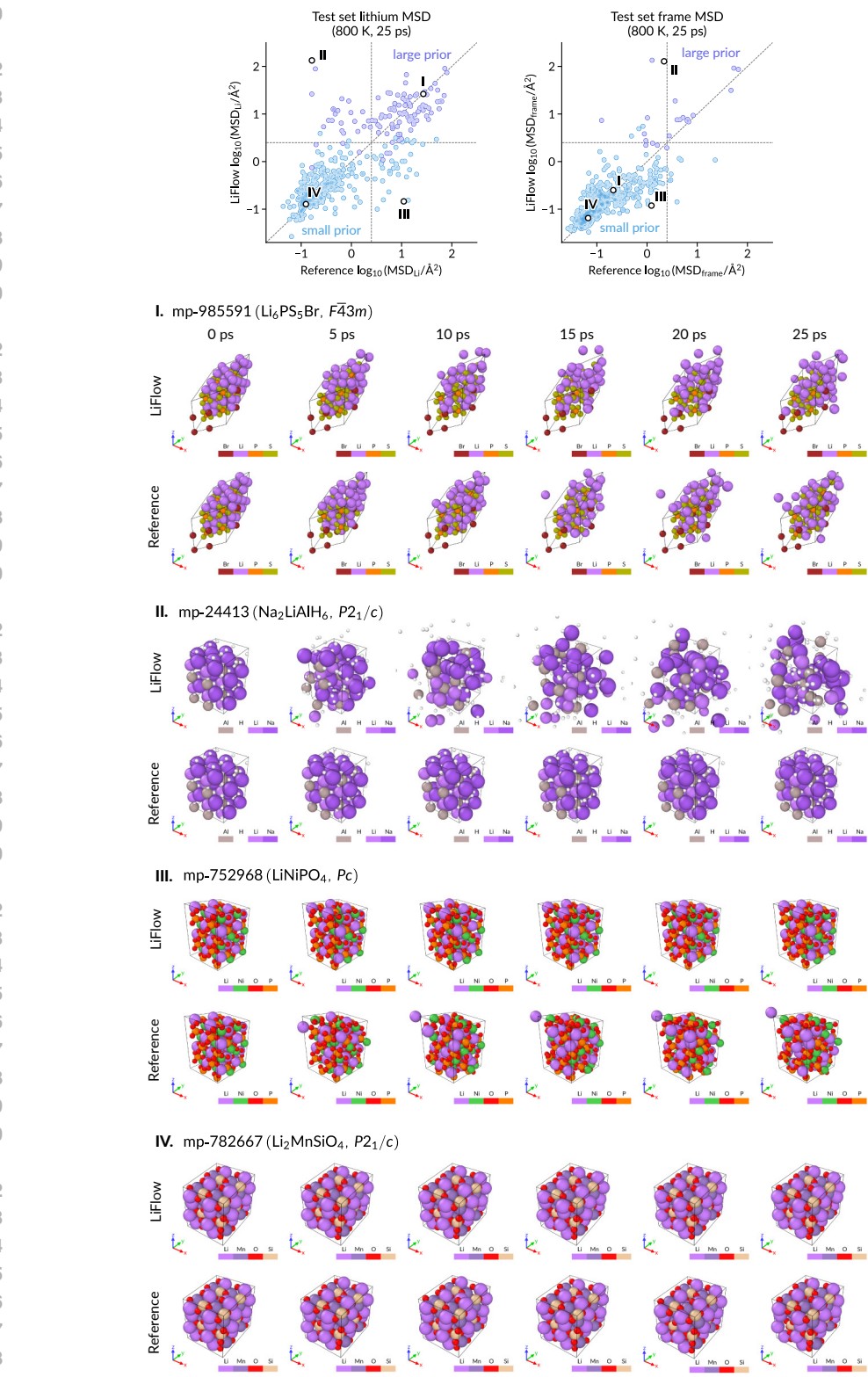

Figure A4: **Universal model inference example.** (Top) Parity plots comparing the log MSD values for lithium and frame atoms in 800 K simulations (reference vs. 25-step LIFLOW inference) across 419 test materials. Data points are colored by their respective prior scales, with four annotated examples (**I–IV**) highlighted below. **II** and **III** represent failed cases where lithium MSD is overestimated and underestimated, respectively. Dotted lines indicate the classification boundary between large and small priors. (Bottom) Reference and generated trajectories for the four annotated test set materials.

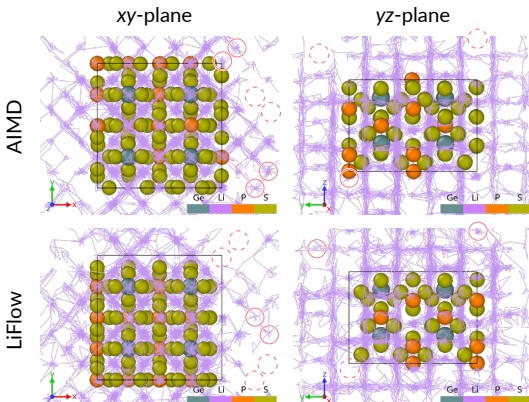

Figure A5: **Diffusion trace of lithium in LGPS simulations.** The diffusion traces of lithium atoms for 150 ps trajectories using LIFLOW and AIMD at 900 K. Different lithium sites are accessed in different simulations, as indicated by circles: solid circles represent sites visited in the current simulation, while dotted circles indicate sites not visited in this simulation but visited in another.

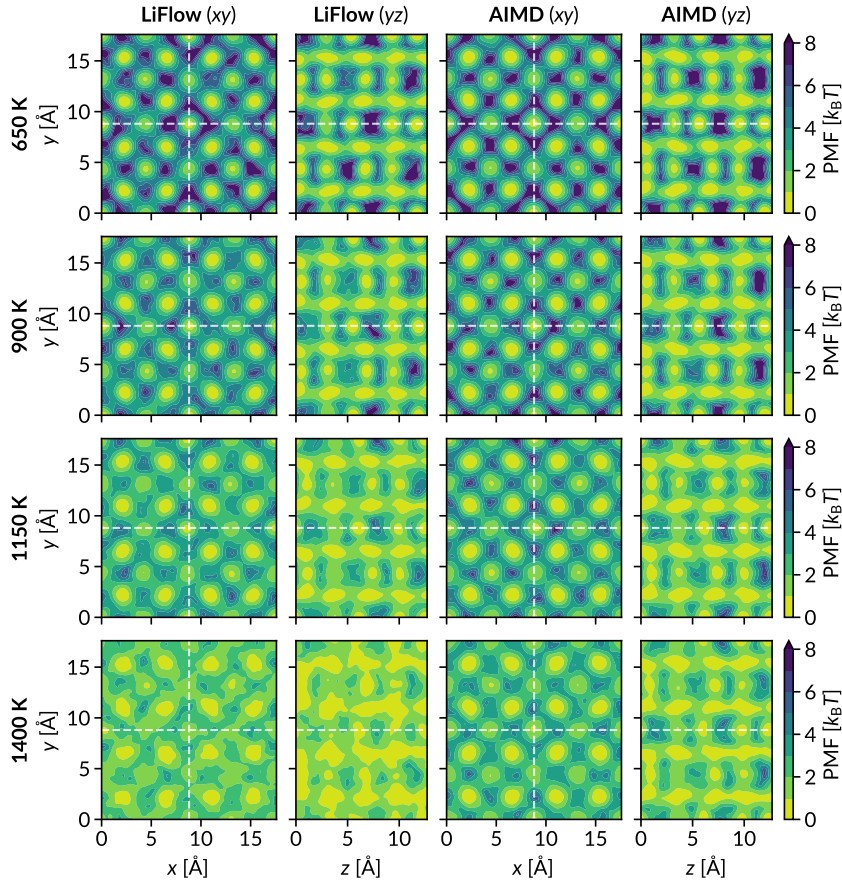

Figure A6: **Negative log densities for lithium in LGPS simulations.** The negative log density $-\log p(\boldsymbol{x})$, or potential of mean force (PMF) in units of $k_\mathrm{B}T$, of lithium atoms in wrapped coordinates is shown for 150 ps trajectories using LIFLOW and AIMD across different temperatures. For each method, the first and second columns correspond to projections along $x$–$y$ and $y$–$z$ planes, respectively. Dotted lines indicate the supercell boundaries.

### E.3 HYPERPARAMETER SENSITIVITY

To evaluate the impact of prior and noise distribution scale hyperparameters on predicting kinetic properties, we perform a sensitivity analysis using the LGPS dataset. For the *Propagator* scales (lithium and frame) and the *Corrector* noise scale, we vary the scales from $\times 1/2$ to $\times 2$, train the corresponding models, and conduct a 150-step (150.75 ps) LiFLOW inference for each model as described in the main text. Results in Fig. A7 demonstrate that diffusivity values show minor deviations from their peak value at the optimal *Propagator* prior scales. Changing *Corrector* noise scale as in Fig. A7c demonstrates that the *Corrector* noise scale larger than a certain threshold causes diffusivities to decrease, suggesting that stronger correction enhances stability but diminishes diffusive behavior slightly.

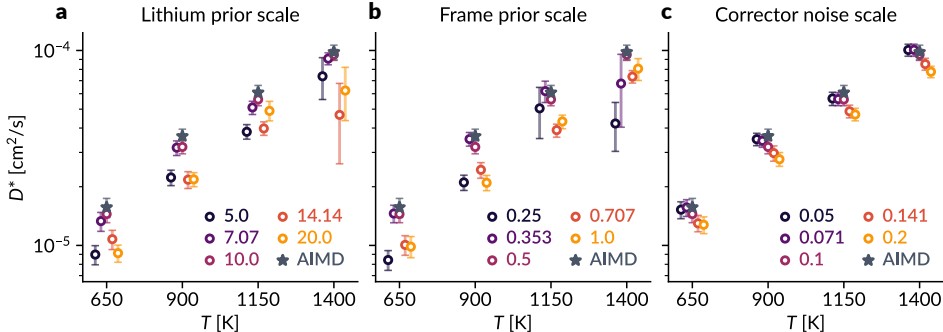

Figure A7: **Scale hyperparameter sensitivity for LGPS models.** **(a)** Variation of the *Propagator* lithium prior scale (default: 10.0). **(b)** Variation of the *Propagator* frame prior scale (default: 0.5). **(c)** Variation of the *Corrector* noise scale (default: 0.1). Results from AIMD reference simulations are also included.

While the *Corrector* significantly improves inference for materials with varying compositions (universal model, Table 1), we found that it plays a reduced role in AIMD models, where training and inference involve the same material structure, as the *Propagator* is sufficiently trained to allow simplified *Corrector* inference. In Fig. A8a and b, we analyze reducing *Corrector* flow steps and performing *Corrector* inference every $n$ *Propagator* steps (e.g., $PPPCPPPC\cdots$ for $n = 3$ versus $PCPCPC\cdots$ for $n = 1$). Diffusivity values remain largely unaffected in both cases. However, when we extend the inference to 1,000 steps (1.005 ns, Fig. A8c), we could observe that higher $n$ values lead to propagation instability at elevated temperatures.

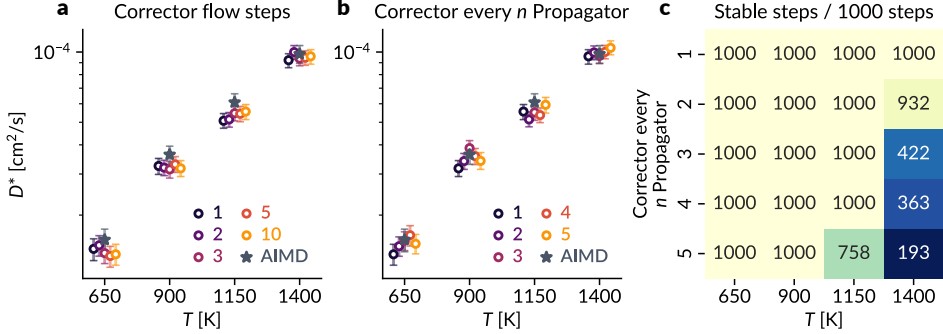

Figure A8: ***Corrector* inference ablation for LGPS models.** **(a)** Variation of the *Corrector* flow steps ($N_{\text{flow}}$, default: 10). **(b)** Applying the *Corrector* every $n$ *Propagator* steps (default: 1). **(c)** Number of stable propagation steps over a 1,000-step inference. Results from AIMD reference simulations are also included.

