# OpenReview forum: "Flow Matching for Accelerated Simulation of Atomic Transport in Materials"
_ICLR.cc/2025/Conference — Submitted to ICLR 2025_

### Official Review · Reviewer_npoR · 2024-10-29

**Soundness:** 2
**Presentation:** 2
**Contribution:** 2
**Rating:** 6
**Confidence:** 1

**Summary:**

As informed to the area chairs, I’m unable to review this paper due my lack of expertise in this area. Please disregard the assigned scores, since I’ve added them just to complete the submission.

**Strengths:**

As informed to the area chairs, I’m unable to review this paper due my lack of expertise in this area. Please disregard the assigned scores, since I’ve added them just to complete the submission.

**Weaknesses:**

As informed to the area chairs, I’m unable to review this paper due my lack of expertise in this area. Please disregard the assigned scores, since I’ve added them just to complete the submission.

**Questions:**

As informed to the area chairs, I’m unable to review this paper due my lack of expertise in this area. Please disregard the assigned scores, since I’ve added them just to complete the submission.

---

> ### Author Response · Authors · 2024-11-23
> **Author comment**
>
> Dear Reviewer npoR,
>
> We appreciate your consideration and letting us know. We wish you the very best with your future research :)
>
> Best regards,
>
> Submission11944 Authors

---

### Official Review · Reviewer_aDcr · 2024-11-01

**Soundness:** 3
**Presentation:** 4
**Contribution:** 2
**Rating:** 5
**Confidence:** 5

**Summary:**

The manuscripts proposes LiFlow, a conditional generative model as a surrogate of MD simulation for Li diffusion in electrolytes, an important technical application. The goal of accelerating MD with generative surrogate model is a rapidly developing field and as the authors noted, there are a good number of papers. There is no substantial and quantitative comparison to those papers. While the technical work here is solid, it lacks enough novelty or substantial improvement as a methodology development paper.

**Strengths:**

The paper was written in a clear, easy to understand way. The technical details and results were clearly presented. Citations were comprehensive and demonstrated knowledge of the authors with the fast-evolving field. The experiments were performed nicely. The results were decent. The release of of an open dataset for Li diffusion simulations is a positive contribution.

**Weaknesses:**

The ideas presented are not very new

* The overall idea of flow-based generative model for MD surrogate modeling was employed in e.g. TimeWarp (Karras et al 2023), and F3LOW (Li et al 2024) even though the applications of these papers were protein dynamics.

* The equivariant GNN (PaiNN) was adopted from an existing, mature design.

* The periodicity aspect is trivial for any work on crystalline materials modeling.

* As the authors properly cited, the predictor (called propagator here) + corrector idea was previously employed in (Fu et al 2023). This paper also discussed Li-ion diffusion.

* The idea of two sets edge in the conditional GNN network was previously employed in (Hsu et al 2024).

* The physics-inspired prior was, as the authors admitted, employed previously.

* The results were good as proof-of-concept, but not practically useful enough.

**Questions:**

* As discussed in the paper, a majority of generative surrogates for MD were applied to biomolecular simulation. A closely related paper, Fu et al 2023, already studied surrogate models based on the diffusion model for Li-ion electrolyte. In their response to this review, the authors are advised to compare more carefully to that paper and show that they are offering something better.

* Please consider explaining how the work compares to TimeWarp and F3LOW.

* Please cite Arts et al 2023, "Two for one ...".

* Overall, I find it hard to accept it as a method development paper to ICLR. It can be a good paper for a chemistry/materials science journal.

---

> ### Author Response · Authors · 2024-11-23
> **Author comments [1/4]**
>
> Dear Reviewer aDcr,
>
> Thank you for your thoughtful review and valuable feedback. We are grateful for your recognition of the clarity and technical presentation. We acknowledge that certain aspects of our approach build on existing methods; however, we believe our work offers novel contributions, such as extending flow-based generative models to atomic transport, integrating physics-inspired priors, and a unique correction mechanism. Additionally, we aim for our work to be understood holistically, encompassing the task requirements, dataset, and proposed methods. Please find our point-by-point response below, and we apologize for any potential errors in our reply due to the limited time available for this rebuttal.
>
> ---
>
> ## Weaknesses
> > The ideas presented are not very new
> >
> > **W1:** The overall idea of flow-based generative model for MD surrogate modeling was employed in e.g. TimeWarp (Karras et al 2023), and F3LOW (Li et al 2024) even though the applications of these papers were protein dynamics.
>
> Our primary contribution is not the general use of flow matching or neural ODEs for MD surrogates in general but their application to atomic transport in crystalline systems by designing a specific task that integrates these methods and introducing novel mechanisms such as a physically motivated adaptive prior. Furthermore, what needs to be modeled depends on specific atomistic system of interest and the target physical property, and previous methods mentioned in the comment are not directly applicable to our specific system.
>
> Timewarp [1] aims to accelerate the sampling over equilibrium conformational states as measured by effective sample size per simulation time. Hence, their propagator is a normalizing flow model that gives exact likelihood as a proposal distribution which is then accepted or rejected by Metropolis-Hastings algorithm. While this ensures that samples are thermodynamically consistent with the Boltzmann distribution, it does not preserve kinetics of the reference dynamics and most of the proposals are rejected during MCMC. Our main objective is reproducing kinetic observables related to atomic transport with faster kinetics at elevated temperatures and mutable chemical environments. Thus, we opted for (1) a rejection-free Corrector mechanism for improved simulation efficiency and (2) flow matching instead of normalizing flows because we could prioritize expressivity over exact likelihoods.
>
> F3low [2] is a relevant flow matching approach working on coarse-grained (CG) backbone frames of proteins. While the same general idea of frame-to-frame generative model is utilized (also as in the previous works such as Timewarp [1] and ITO [3]), the key difference lies in that F3low (1) uses "conformation guidance" mechanism to model conditional time-propagation distributions of positions. While their choice of prior distribution is isotropic Gaussian (normal for translations and IGSO(3) for rotations), they incorporated the conditional previous time step structure by interpolating it with the prior sample on respective manifolds. This is incompatible with translational symmetries in crystals, as interpolated positions shift to different crystallographic positions. Also, (2) proteins benefit from canonical CG backbone representations on SE(3), while crystalline systems studied in this work lack such representations due to mutable bonding and the absence of well-defined, time-invariant chemical entities. Although we suppose that incorporating such inductive biases could enhance the chemical transferability of generative dynamics, F3low has only been demonstrated on identical training and test systems from a commonly used set of fast-folding proteins.
>
> Finally, we want to highlight that, unlike the isotropic Gaussian priors used in both methods, we developed an adaptive Maxwell-Boltzmann prior that is physically motivated, composition- and temperature-dependent, and empirically beneficial, as demonstrated in our ablation studies.
>
> > **W2:** The equivariant GNN (PaiNN) was adopted from an existing, mature design.
> >
> > **W5:** The idea of two sets edge in the conditional GNN network was previously employed in (Hsu et al 2024).
>
> We do not claim novelty in the PaiNN [4] architecture or the introduced modifications. Our focus is on designing the task and defining the requirements, enabling the use of any architecture that can process two sets of positional information (L = 1) and output a vector field for displacements (L = 1). We chose PaiNN for its fast inference speed and reasonable benchmarking performance in ML interatomic potential literature. Our contributions lie elsewhere, as mentioned previously.

---

> ### Author Response · Authors · 2024-11-23
> **Author comments [2/4]**
>
> Furthermore, we explicitly acknowledged the use of two edge sets in Appendix D.2 ("Message Passing") with the note: "A similar approach using two sets of edge information was previously employed by Hsu et al. (2024)." [5] Since we do not claim novelty in architectural modifications, we want to emphasize that this section compares empirical choices for incorporating intermediate displacement information. While the original PaiNN model is theoretically capable of handling displacement inputs (vector input features), we investigate how these inputs are best utilized in our set of experiments.
>
> > **W3:** The periodicity aspect is trivial for any work on crystalline materials modeling.
>
> While we agree that any work on crystalline materials must account for periodicity, we disagree that it is trivial in the context of our work. Periodicity impacts two key aspects: (1) material representation and (2) task design for generative models. The first aspect involves incorporating lattice matrices and constructing edges across unit cell boundaries, which is indeed straightforward, as acknowledged by citations of seminal works in Section 2.1. However, the second aspect, task design, is non-trivial. Previous generative models for materials have focused on generating atoms within periodic boundaries, where atoms potentially cross these boundaries during the generative process. Directly applying this scheme to model the conditional distribution of future positions risks losing track of atomic transport over long time scales (detailed in Appendix A.3). To address this, we model in unwrapped positions while enabling periodic interactions. This approach aligns more closely with how ML interaction potentials for materials operate, rather than previous generative models for crystalline structures. Since this connection between energy/force models and generative models has not yet been clearly established in the materials ML literature, and we aim to clarify this distinction.
>
> > **W4:** As the authors properly cited, the predictor (called propagator here) + corrector idea was previously employed in (Fu et al 2023). This paper also discussed Li-ion diffusion.
>
> As mentioned in the main text, our introduction of Corrector is inspired by Fu et al. [6], but their operation and training are different due to variations in modeling schemes.
>
> Since the Fu et al. [6] employs a CG representation of atomistic systems, irrelevant high frequency movements in the system are integrated out, and the long-term dynamics could be modeled by an equation of motion in non-Markovian sense, for which acceleration is predicted using graph neural networks. Hence, their corrector could be trained to align the predicted positions with reference positions, leveraging the assumption that, given previous frames, the next position is close to the reference simulation.
>
> In contrast, since we cannot assume a CG representation as previously mentioned, we instead adopt a Markovian modeling for all-atom dynamics, using a time scale at which velocity autocorrelation is sufficiently small. Hence, there would be a distribution of potential next frame structures in our case, and mapping the generated structure or displacement into "true" one would be ill-defined because there is no single "true" reference in our case. Furthermore, if we want to compute some form of loss function using the final output of flow propagation, we would have to backpropagate through the flow, which would be computationally infeasible. Hence, our Corrector model is instead decoupled from the Propagator model and is trained to denoise an small-scale arbitrary positional noise. This is also beneficial because Propagator and Corrector could be trained in parallel if computational resources allow.
>
> Also, while based on the idea of the predictor-corrector scheme, we name the model Propagator to avoid confusion with ODE sampling methods and connect the modeling approach to the operator it represents.
>
> > (**W5** is attached to **W2**)
> >
> > **W6:** The physics-inspired prior was, as the authors admitted, employed previously.
>
> Different systems require different priors, and our adaptive prior is the first to be inspired by dynamics. Specifically, it accounts for temperature, mass, and phase dependencies of atomic displacements. The priors discussed in the related works we cited are not designed for models that propagate with accurate dynamics but rather for enhancing the generation of static structures. While both our approach and theirs aim to align the prior distribution more closely with the data distribution to make learning easier, it would be an unfair critique to claim a lack of novelty in our work simply because we share this broad objective and the use of non-Gaussian prior.

---

> ### Author Response · Authors · 2024-11-23
> **Author comments [3/4]**
>
> Specifically, here we examine the previous related works mentioned in the main text here. The decomposed prior proposed by Guan et al. [7] is tailored for design purposes. Their method classifies atom clusters into components such as arms and scaffolds, assigning corresponding priors based on biochemical heuristics rather than physical laws. Similarly, Irwin et al. [8] developed a prior that varies noise scales based on the number of atoms for molecular conformer generation, but this prior also does not consider dynamics. Jing et al. [9] introduced a harmonic prior for protein structure generation, inspired by the harmonic normal modes analysis. This approach, however, applies to a completely different system and does not incorporate additional conditions such as temperature, mass, or phase dependencies, which are central to our method.
>
> > **W7:** The results were good as proof-of-concept, but not practically useful enough.
>
> The practical usefulness of the MD surrogate models would depend on the specific use cases. In our work, given that these models are even more cost-effective than MLIP-based simulations, we envision their application in high-throughput screening of millions of candidate materials to identify those with high diffusivity. In our experiments, we have shown that the model can successfully reproduce the superionic conduction behavior of known lithium-ion conductors like argyrodite (Li₆PS₅Br) and LGPS (Li₁₀GeP₂S₁₂), without requiring training on the same compositions. While we have not demonstrated practical material screening in this study and it is beyond its scope, future work could focus on screening across different compositions, ionic concentrations, and orderings, further advancing the methodology for such applications.
>
> ## Questions
>
> > **Q1:** As discussed in the paper, a majority of generative surrogates for MD were applied to biomolecular simulation. A closely related paper, Fu et al 2023, already studied surrogate models based on the diffusion model for Li-ion electrolyte. In their response to this review, the authors are advised to compare more carefully to that paper and show that they are offering something better.
>
> We would like to clarify that although both Fu et al. [6] and this work involve the same diffusing ionic species (lithium cation), the differences in chemical environments—polymer vs. solid frameworks—require distinct representations of atomic systems and, consequently, different modeling schemes for dynamics. The modeling approach in [6] is not directly applicable to our systems due to the use of a CG representation and time-invariant bond features for graph construction.
>
> As mentioned in previous replies, the CG representation in [6] allows modeling long-term dynamics using a non-Markovian equation of motion. This enables training an acceleration predictor and corrector model with several previous frames as inputs. However, in crystalline systems at higher temperatures, the bonding is mutable, and time-invariant chemical entities over different compositions are challenging to identify. Therefore, a universal CG scheme is not feasible for our systems. Instead, we train a generative model for the propagator to approximate Markovian transitions and use the Corrector mechanism decoupled from the Propagator training.
>
> Additionally, [6] employs time-invariant bond features for graph construction, which is effective for modeling ionic transport in polymeric systems, where ionic movement occurs between coordination environments composed of donor atoms in the polymer and counteranions. In contrast, although time-invariant species could be identified on a case-by-case basis for specific materials, it is practically impossible to find such representations that work across a wide range of materials. Therefore, [6] and our work are tailored to different types of chemical systems, requiring distinct representations and modeling approaches. It would be unjustified to claim that our method is better, but rather both methods introduce specialized approaches for their respective systems of interest.
>
> > **Q2:** Please consider explaining how the work compares to TimeWarp and F3LOW.
>
> Please refer to our comments in **W1** (in Author comments 1).
>
> > **Q3:** Please cite Arts et al 2023, "Two for one ...".
>
> We added Arts et al. [10] to the “ML surrogates for dynamics simulation” paragraph in Sec. 2.2 (Related Works).

---

> > ### Author Response · Authors · 2024-11-23
> > **Author comments [4/4]**
> >
> > > **Q4:** Overall, I find it hard to accept it as a method development paper to ICLR. It can be a good paper for a chemistry/materials science journal.
> >
> > While we highlight our unique contributions in methodology: (1) a physically motivated prior distribution integrated with flow matching, and (2) a corrector model decoupled from propagator, we also want to emphasize that this work is not only a "method paper". It also introduces the task of modeling atomic transport as a generative task, with an appropriate dataset for training and testing models, and provides a reasonable (if not immediately practical) approach with unique task-informed components. Our consideration when submitting this to ML venue instead of chemistry/materials venue was based on the idea that, as noted in the "Limitations and future directions" paragraph in Conclusion, this line of research would benefit more from the methodological development at this stage than from the application perspective. In the context of surrogate models for MD literature, we hope that by framing our task with its methodological contributions and requirements, this work will help contextualize the challenge as an interesting inference/generative modeling task. It offers extended degrees of freedom, such as chemical/thermal transferability, compared to well-established biomolecular simulations using classical force fields in the practical realm.
> >
> > ### References
> >
> > [1] Klein et al., Timewarp: Transferable acceleration of molecular dynamics by learning time-coarsened dynamics. Advances in Neural Information Processing Systems, volume 36, pp. 52863–52883, 2023.
> >
> > [2] Li et al., F3low: Frame-to-frame coarse-grained molecular dynamics with SE(3) guided flow matching, 2024. URL https://arxiv.org/abs/2405.00751
> >
> > [3] Schreiner et al., Implicit transfer operator learning: Multiple time-resolution models for molecular dynamics. Advances in Neural Information Processing Systems, volume 36, pp. 36449–36462, 2023.
> >
> > [4] Schütt et al., Equivariant message passing for the prediction of tensorial properties and molecular spectra. Proceedings of the 38th International Conference on Machine Learning, volume 139, pp. 9377–9388, 2021.
> >
> > [5] Hsu et al., Score dynamics: Scaling molecular dynamics with picoseconds time steps via conditional diffusion model. J. Chem. Theory Comput., 20(6):2335–2348, 2024.
> >
> > [6] Fu et al., Simulate time-integrated coarse-grained molecular dynamics with multi-scale graph networks. Transactions on Machine Learning Research, 2023.
> >
> > [7] Guan et al., DecompDiff: Diffusion models with decomposed priors for structure-based drug design, Proceedings of the 40th International Conference on Machine Learning, volume 202, pp. 11827–11846, 2023.
> >
> > [8] Irwin et al., Efficient 3d molecular generation with flow matching and scale optimal transport, 2024. URL https://arxiv.org/abs/2406.07266
> >
> > [9] Jing et al., EigenFold: Generative protein structure prediction with diffusion models, 2023. URL https://arxiv.org/abs/2304.02198
> >
> > [10] Arts et al., Two for one: Diffusion models and force fields for coarse-grained molecular dynamics. J. Chem. Theory Comput., 19(18):6151–6159, 2023.
> >
> > ---
> >
> > Thank you again for your time and effort in reviewing our work, and please let us know if you have any further suggestions or questions.
> >
> > Best regards,
> >
> > Submission11944 Authors

---

> > > ### Comment · Reviewer_aDcr · 2024-11-25
> > >
> > > I appreciate the careful and detailed response of the authors. As I said in the first-round review, it is a good paper. It was well written and I do not see much weakness in it but I do not see sufficient pushing of the envelope to accept it either as a method paper or as a scientific-ML applications paper for ICLR. I am keeping my score.

---

> > > > ### Author Response · Authors · 2024-11-26
> > > > **Thank you and possible further consideration points**
> > > >
> > > > Dear Reviewer aDcr,
> > > >
> > > > Thank you for your response. We appreciate your positive comments on the quality and clarity of the paper, as well as your acknowledgment of our detailed reply.
> > > > While we respect your decision to maintain the score, we would like to highlight the following points for possible further consideration:
> > > >
> > > > - Our paper introduces unique methodological contributions: (1) a physically motivated prior distribution integrated with flow matching, and (2) a corrector model decoupled from the propagator, as discussed in our comparison to previous studies in the rebuttal.
> > > > - It frames atomic transport modeling as a generative task, presenting challenges distinct from previous work on accelerating MD simulations, as highlighted in our comparison to other methods (thermodynamic sampling, CG modeling, ...) in the rebuttal.
> > > > - As a scientific-ML application, the introduced task offers broader implications for surrogate models in MD, featuring extended control variables and compositional complexity compared to previous work in the field.
> > > >
> > > > Once again, we thank you for your time and consideration.
> > > >
> > > > Thank you,
> > > >
> > > > Submission11944 Authors

---

### Official Review · Reviewer_VS8V · 2024-11-01

**Soundness:** 3
**Presentation:** 2
**Contribution:** 3
**Rating:** 8
**Confidence:** 4

**Summary:**

This paper proposes a machine-learning-accelerated approach to simulating atomic dynamics in crystalline materials. The approach uses a flow matching model to predict the atomic positions at the next (physical) time given their current positions. An auxiliary “corrector” flow matching model denoises the predicted displacements of any errors introduced by the aforementioned “propagator” model. The authors take care to design an equivariant flow and invariant prior distribution that respect symmetries inherent in the problem. The authors created two datasets of MLIP and AIMD computed trajectories, respectively, to train the flow matching models. They compare the mean squared displacement (MSD) and radial distribution functions (RDF) of the estimated trajectories against those of the ground truth. Ablation studies verify the design choices.

**Strengths:**

* The proposed method is sound, and the ablation studies show significant improvement provided by each of the components (i.e., the task-informed prior and corrector model).
* I especially like that the authors elegantly incorporate physics knowledge in the generative approach. Namely, they choose a prior based on the Maxwell-Boltzmann distribution, check that the prior distribution is invariant with respect to the relevant symmetries, and check that the flow (although linear interpolation is already a common choice) is equivariant with respect to the symmetries.
* Overall, the presentation is good. The background was easy to follow and provided just enough information for a non-domain-expert to understand the paper. The reason I rate the presentation as a 2 instead of a 3 is because the results section refers too much to the appendix, and technically the paper (i.e., everything before references) goes beyond ten pages.

**Weaknesses:**

* The MSD and RDF metrics appear to be high-level summary metrics of an estimated trajectory, making me wonder how much information is lost when using them to compare to the ground truth. For example, why is RDF not averaged across the whole simulation (line 345)? Why not compute something like MAE of estimated positions across the whole trajectory?
* The authors should have made more effort to make the presentation compact enough to fit on 9-10 pages without having to refer to appendix figures. All the material up to the references technically takes up more than 10 pages. As an example of excessive reliance on appendix material, an entire subsection (lines 455-466) refers only to appendix figures.

**Questions:**

* Intuitively, it seems that the corrector model should be easier to learn than the propagator model since its task is simpler (just removing small amounts of noise from displacements). Why not make it a one-step conditional generative model instead of an entire flow? Have you tried reducing the number of flow steps to 1 for the corrector?

Suggestions:
* Line 71 mentions a “lattice matrix.” It would help to describe what this is intuitively for someone unfamiliar with the field.
* In the footnote on line 107, should the $t$ be a $\tau$?
* Line 230 defines a $\sigma_\mathcal{S}$ variable that does not appear in Equation 10 above.
* The language in lines 331-332 makes it sound like the MSD is computed *between* configurations and not for a single configuration.
* It’s hard to tell what the purpose of Table 3 is, and it’s not explained much in the text. What are these results supposed to convey about your method?

---

> ### Author Response · Authors · 2024-11-23
> **Author comments [1/2]**
>
> Dear Reviewer VS8V,
>
> Thank you for your detailed and constructive review. We appreciate your positive feedback on the soundness of our approach, the integration of physics knowledge, and the overall presentation of our work. In response to your comments, we have significantly revised the Results section to reduce reliance on the Appendix. Additionally, we have incorporated your idea of simplifying the Corrector inference and included additional ablation studies in Appendix E.3. Please find our point-by-point response below, and we apologize for any potential errors in our reply due to the limited time available for this rebuttal.
>
> ---
>
> ## Weaknesses
>
> > **W1:** The MSD and RDF metrics appear to be high-level summary metrics of an estimated trajectory, making me wonder how much information is lost when using them to compare to the ground truth. For example, why is RDF not averaged across the whole simulation (line 345)? Why not compute something like MAE of estimated positions across the whole trajectory?
>
> We average the RDF over the latter 80% (20 ps) of the simulation in the universal model experiments, discarding the first 20% (5 ps) to allow for equilibration. While this does not cover the entire simulation, it captures the equilibrated portion, a major part of the simulation that better reflects the equilibrium statistics. MAE of estimated positions was not applicable here due to the stochastic nature of both the reference dynamics and the generated trajectories, driven by random initial velocities and prior sampling, respectively.
>
> Recognizing that MSD and RDF are high-level summary metrics, we have also included additional qualitative and quantitative assessments: visual inspection of simulation trajectories (universal model, Fig. 2 and A4) and lithium diffusion traces and lithium position probability densities (AIMD model, Figs. A5 and A6) to assess structural feature reconstruction.
>
> > **W2:** The authors should have made more effort to make the presentation compact enough to fit on 9-10 pages without having to refer to appendix figures. All the material up to the references technically takes up more than 10 pages. As an example of excessive reliance on appendix material, an entire subsection (lines 455-466) refers only to appendix figures.
>
> Thank you for pointing out the over-reliance on Appendix materials in the main text, and we have accordingly revised the entire Experiments section (Sec. 4) to streamline the exposition of results. Specifically, we identified that Fig. A4 (Universal model inference example) was used to discuss universal model results in Sec. 4.2. To address this, we extracted the key elements from Fig. A4 and created a new figure (Fig. 2) in the main text, under the paragraph “Reproducing kinetic properties” in Sec. 4.2, to ensure the discussion in the main text remains self-contained. Additionally, as the reviewer noted, the paragraph “Reproducing structural features” referred entirely to Appendix figures (Figs. A5 and A6). In the revised version, we clarified that this section is supplemental to the main results and deferred detailed discussions to Appendix E.2, while providing a summarized version in the main text. We hope the revised manuscript offers a cleaner and more concise presentation.
>
> ## Questions
>
> > **Q1**: Intuitively, it seems that the corrector model should be easier to learn than the propagator model since its task is simpler (just removing small amounts of noise from displacements). Why not make it a one-step conditional generative model instead of an entire flow? Have you tried reducing the number of flow steps to 1 for the corrector?
>
> We agree with your intuition and conducted additional ablation studies to confirm this, as detailed in Appendix E.3. Compared to the universal model, where the Corrector is highly beneficial (Table 1), training and inference on AIMD models are conducted on materials with the same composition, resulting in less pronounced Propagator errors. Additional experiments examined simplifying Corrector inference by reducing the number of flow integration steps or applying the Corrector intermittently across Propagator steps for LGPS AIMD models (Appendix E.3, Fig. A8). Results showed that while these simplifications do not significantly affect diffusivity values in 150-step LGPS inference, the Corrector remains critical for extending dynamics up to 1,000-step inference at higher temperatures.
>
> > **Q2:** Suggestions:
> > - Line 71 mentions a “lattice matrix.” It would help to describe what this is intuitively for someone unfamiliar with the field.
> We added explanation to the lattice matrix:
> * $L = (l_1, l_2, l_3)^\top \in \mathbb{R}^{3 \times 3}$ is the lattice matrix *with rows defining the basis vectors of a 3-D repeating unit cell,* ...

---

> > ### Author Response · Authors · 2024-11-23
> > **Author comments [2/2]**
> >
> > > - In the footnote on line 107, should the $t$ be a $\tau$?
> >
> > Displacement indices for $D_0$ and $D_1$ represent the flow matching times, corresponding to $t = 0$ and $1$, respectively. Since the physical time for displacements is always $\Delta \tau$, it is omitted.
> >
> > > - Line 230 defines a $\sigma_\mathcal{S}$ variable that does not appear in Equation 10 above.
> >
> > We revised the part to avoid undefined variables.
> > - (Before) where $\sigma_\mathcal{S}$ selects a scale value, either $\sigma_\mathcal{S}^\text{small}$ or $\sigma_\mathcal{S}^\text{large}$ (both hyperparameters), based on the output of a binary classifier that predicts whether the displacements for species $\mathcal{S}$ (lithium or frame) will be small or large.
> > - (After) where for each species $\mathcal{S} \in$ {lithium, frame}, $\sigma_\mathcal{S}$ selects a scale value from the hyperparameters {$ \sigma_\mathcal{S}^\text{small}, \sigma_\mathcal{S}^\text{large} $} based on a binary classifier's prediction of whether the displacements for $\mathcal{S}$ will be small or large.
> >
> > > - The language in lines 331-332 makes it sound like the MSD is computed between configurations and not for a single configuration.
> >
> > We removed the phrase “to the reference trajectories” to make it clear that they are not computed between configurations.
> > * (Before) Given the wide range of magnitudes in the raw MSD values, we compared the log MSD values (with base 10, MSD in units of Å$^2$) to the reference trajectories.
> > * (After) Given the wide range of magnitudes of MSD values, we compared the log values (base 10) of MSD, with MSD in units of Å$^2$.
> >
> >
> > > - It’s hard to tell what the purpose of Table 3 is, and it’s not explained much in the text. What are these results supposed to convey about your method?
> >
> > We noticed that Table 3 in the original version was placed far from the section where its results were discussed. The results in Table 3 pertain to the characterization of the temperature dependence of diffusivity values from each trajectory, including activation energies and their confidence intervals, as well as confirming that LiFlow is consistent with the reference trajectory results. As part of the reorganization of Sec. 4, we moved Table 3 (now Table 2) directly next to the discussion in the "Reproducing kinetic properties" paragraph in Sec. 4.3.
> >
> > ---
> >
> > Thank you again for your time and effort in reviewing our work, and please let us know if you have any further suggestions or questions.
> >
> > Best regards,
> >
> > Submission11944 Authors

---

> > > ### Comment · Reviewer_VS8V · 2024-11-26
> > >
> > > Thank you to the authors for their efforts in the rebuttal. Figure A8 suggests that a 1-step Corrector model is sufficient, in which case I suggest making that the default setting in the main paper or at least mentioning that a 1-step Corrector may be sufficient. The authors have addressed most of my questions, so I will raise my rating.

---

> > > > ### Author Response · Authors · 2024-11-26
> > > > **Thank you**
> > > >
> > > > Dear Reviewer VS8V,
> > > >
> > > > Thank you for your positive feedback and evaluation of our work. We have additionally incorporated the use of the 1-step Corrector for AIMD models in Sec. 4.3 (paragraph "Setup") and Appendix D.4 as follows:
> > > >
> > > > - (Sec. 4.3) Corrector inference can also be simplified by reducing $N_\text{flow}$, as detailed in Appendix E.3.
> > > > - (Appendix D.4) For AIMD simulations, since the Propagator error is relatively small, Corrector inference can be simplified without impacting simulation results—for example, by reducing $N_\text{flow}$ to $1$. Details of these ablation studies are provided in Appendix E.3.
> > > >
> > > > We sincerely appreciate your suggestion, which provided a promising direction for enhancing computational efficiency. We look forward to acknowledging this in the final version of the manuscript.
> > > >
> > > > Thank you,
> > > >
> > > > Submission11944 Authors

---

### Official Review · Reviewer_xeRn · 2024-11-03

**Soundness:** 4
**Presentation:** 3
**Contribution:** 4
**Rating:** 8
**Confidence:** 3

**Summary:**

The authors introduce a generative modeling framework called LIFlow that serves to accelerate molecular dynamics (MD) simulations of crystalline materials. A physically-correct propagator is combined with a stabilizing corrector, and an adaptive prior based on the Maxwell-Boltzmann distribution is leveraged. The resulting generative model is thoroughly evaluated and results in massive speedups over direct MD simulations, all while maintaining high accuracy.

**Strengths:**

The authors introduce a novel method for generative acceleration of molecular dynamics (MD) simulations for crystalline materials by generating atomic displacements. They also develop a flow matching approach that takes into account chemical and thermal conditions, and incorporate a corrector to ensure stability. The authors also contribute a new dataset useful for the material science community. Overall, the paper is exceptionally well written and the methods appear sound and novel, to the best of my knowledge. As written, the scope feels laser focused on MD simulations, but I'm confident the ideas in this work could be applied to other scenarios as well; see the weaknesses and questions sections for some ideas there.

**Weaknesses:**

The primary issue is that the paper needs to be cleaned up for more general audiences, by including more citations where appropriate and adding more information to the appendix section. See "Questions" for more concrete details and suggestions. I encourage the authors to revise the paper so a broader audience can read and understand what's going on, not just folks from the atomic transport and SSE community.

**Questions:**

1. I had to do a lot of background reading on atomic transport in order to even begin understanding this paper. I suggest that the authors include a set of links to seminal works, right at the end of the first sentence for the paper.

2. I understand space is at a premium, but the paragraph on page 4, lines 169-176, is very dense and assumes a lot of prior knowledge with no citations. Please add citations so the interested reader can follow them. ICLR is a broad conference for a general audience, and it is plausible that your techniques may be usable elsewhere.

3. Eq 11, why linear interpolation? Is the only rationale to come up with a simple combination of the prior sample and the data sample to design the flow + also satisfy symmetry conditions?

4. On line 245 and 246 on page 5, you say you use an RBF expansion of atomic distances and the unit vector directions along edges. This is one of those examples of an extremely dense statement that should be unpacked somewhere (perhaps the appendix). Please elaborate.

5. Please explain your evaluation metrics more clearly for someone who is not from your community. I suggest adding a section to the appendix. I had to do a lot of searching to figure out why your presented metrics were reasonable ones.

---

> ### Author Response · Authors · 2024-11-23
> **Author comments [1/2]**
>
> Dear Reviewer xeRn,
>
> Thank you for your thoughtful and thorough review of our work. We appreciate your recognition of the novelty and soundness of our methods, as well as the contribution of our new dataset. In response to your feedback, we have revised the manuscript to better address a broader audience, incorporating additional citations. Furthermore, we have improved the exposition of the Results section to enhance readability. Please find our point-by-point response below, and we apologize for any potential errors in our reply due to the limited time available for this rebuttal.
>
> ---
>
> ## Weaknesses
>
> > **W1:** The primary issue is that the paper needs to be cleaned up for more general audiences, by including more citations where appropriate and adding more information to the appendix section. See "Questions" for more concrete details and suggestions. I encourage the authors to revise the paper so a broader audience can read and understand what's going on, not just folks from the atomic transport and SSE community.
>
> Thank you for your thorough review and for engaging with a topic that may be outside your primary area of familiarity. We greatly appreciate your detailed suggestions and have revised the manuscript to make it more accessible to a broader audience.
>
> ## Questions
>
> > **Q1:** I had to do a lot of background reading on atomic transport in order to even begin understanding this paper. I suggest that the authors include a set of links to seminal works, right at the end of the first sentence for the paper.
>
> We have addressed this by adding references to a general introductory textbook on kinetics and atomic transport in materials [1] and a general introduction to atomistic simulations in materials science [2] at the end of the first sentence of the Introduction.
>
> > **Q2:** I understand space is at a premium, but the paragraph on page 4, lines 169-176, is very dense and assumes a lot of prior knowledge with no citations. Please add citations so the interested reader can follow them. ICLR is a broad conference for a general audience, and it is plausible that your techniques may be usable elsewhere.
>
> Thank you for bringing this to our attention. To address this, we have added a citation for usual MD time step of 1 fs [3] and for the use of unwrapped coordinates in kinetics simulations [4]. We hope these additions will help interested readers better understand and contextualize the discussion.
>
> > **Q3:** Eq 11, why linear interpolation? Is the only rationale to come up with a simple combination of the prior sample and the data sample to design the flow + also satisfy symmetry conditions?
>
> It is indeed true that linear interpolation between the prior and data samples is not the only method to satisfy the symmetry conditions. During the development of this method, we also considered displacing atoms along a geodesic on a Riemannian manifold to prevent collisions during flow propagation. While this approach satisfied the symmetry conditions and provided reasonable (if not optimal) results, we found that introducing shifts to the distances before embedding them effectively resolved the same issue. As a result, we adopt the conditional optimal transport path because of its computational efficiency.
>
> > **Q4:** On line 245 and 246 on page 5, you say you use an RBF expansion of atomic distances and the unit vector directions along edges. This is one of those examples of an extremely dense statement that should be unpacked somewhere (perhaps the appendix). Please elaborate.
>
> While this approach is commonly employed in equivariant graph neural networks, in our work, it is specifically inherited from the PaiNN model [5] that we utilize. The use of distances and unit vectors in message passing is detailed in Appendix D.2, particularly in Eqs. (26a) and (27a). To address this concern, we have added a reference to those in the main text to direct readers to the relevant explanation.
>
> > **Q5:** Please explain your evaluation metrics more clearly for someone who is not from your community. I suggest adding a section to the appendix. I had to do a lot of searching to figure out why your presented metrics were reasonable ones.
>
> We have added intuitive explanations for each evaluation metric before their formal definitions in the main text:
>
> * "The MSD measures the average squared distance that particles of type $\mathcal{S}$ move over time $\tau$, ..."
> * "... the RDF describes how particle density varies as a function of distance from a reference particle, revealing spatial organization and local structure in the system, and defined as: ..."

---

> > ### Author Response · Authors · 2024-11-23
> > **Author comments [2/2]**
> >
> > Furthermore, while we have not included extended details in the Appendix, we have identified a publication in an ML venue benchmarking MLIP-based simulations [6], which provides a detailed description of a similar set of metrics. We have added a note with a citation to this work to provide additional context. While keeping the details concise, we hope the revised explanations make the metrics more accessible. We remain open to further elaboration if needed and hope this enhances the clarity of the manuscript.
> >
> > ### References
> > [1] Balluffi et al., Kinetics of Materials. John Wiley & Sons, 2005.
> >
> > [2] Sidney Yip, Molecular Mechanisms in Materials: Insights from Atomistic Modeling and Simulation. MIT Press, 2023.
> >
> > [3] Marx and Hutter. Ab Initio Molecular Dynamics: Basic Theory and Advanced Methods. Cambridge University Press, 2009.
> >
> > [4] von Bülow et al., Systematic errors in diffusion coefficients from long-time molecular dynamics simulations at constant pressure. J. Chem. Phys., 153(2):021101, 2020.
> >
> > [5] Schütt et al., Equivariant message passing for the prediction of tensorial properties and molecular spectra. Proceedings of the 38th International Conference on Machine Learning, volume 139, pp. 9377–9388, 2021.
> >
> > [6] Fu et al., Forces are not enough: Benchmark and critical evaluation for machine learning force fields with molecular simulations. Transactions on Machine Learning Research, 2023.
> >
> >
> > ---
> > Thank you again for your time and effort in reviewing our work, and please let us know if you have any further suggestions or questions.
> >
> > Best regards,
> >
> > Submission11944 Authors

---

> > > ### Comment · Reviewer_xeRn · 2024-11-25
> > >
> > > Thanks to the authors for responding. All sounds good.

---

> > > > ### Author Response · Authors · 2024-11-26
> > > > **Thank you**
> > > >
> > > > Dear Reviewer xeRn,
> > > >
> > > > Thank you for reviewing our comments and providing your response. We appreciate your encouraging feedback.
> > > >
> > > > Thank you,
> > > >
> > > > Submission11944 Authors

---

### Official Review · Reviewer_Z7DJ · 2024-11-03

**Soundness:** 3
**Presentation:** 3
**Contribution:** 2
**Rating:** 3
**Confidence:** 5

**Summary:**

The paper introduces LIFLOW, a generative framework for improving molecular dynamics (MD) simulations in crystalline materials, particularly solid-state electrolytes (SSEs). It formulates the simulation task as conditional generation of atomic displacements and employs a flow matching approach with a 'Propagator' for generating atomic displacements and a 'Corrector' to ensure physical stability. Model also leverages adaptive prior based on the Maxwell–Boltzmann distribution to account for chemical and thermal conditions. The model is benchmarked on a dataset of lithium diffusion across SSE candidates, achieving significant speed-ups over traditional Ab Initio Molecular Dynamics (AIMD) methods.

**Strengths:**

The application of flow matching to model atomic transport is an innovative approach, offering a transformative potential for simulating dynamics efficiently across large spatiotemporal scales. LIFLOW achieves remarkable computational performance, delivering speed-ups of up to 600,000× compared to AIMD and 400× compared to MLIP-based simulations. By tackling the significant challenge of high computational costs in MD simulations, this work addresses a crucial bottleneck in materials research. The focus on lithium-based solid-state electrolytes is both timely and highly relevant, given the growing need for advancements in energy storage technologies.

**Weaknesses:**

The primary comparison is with AIMD simulations and MLIP, which are used to generate the data. The paper does not compare LIFLOW to existing machine learning-based approaches for MD acceleration, such as GNN-based models [1, 2], diffusion models [3], variational approaches [4] or normalizing flows [5]. Inclusion of such comparisons would greatly improve the credibility of the claims of ths work.

While the model is shown to work well on lithium SSEs, there is limited discussion on generalizing LIFLOW to other types of materials or chemical systems. This could restrict the perceived impact of the method. An additional showcase on different datasets, such as MD17 [6] and OC20 [7] could further shine light on the model's generalization.

The model struggles with extrapolating to conditions outside the training regime, such as lower temperatures, where rare events are poorly captured. This suggests potential limitations in scenarios requiring greater generalization. Considering well know enhancement approaches, such as physics informed neural networks for the loss function modification or active learning to sample when rare events occur can be benificial.

The model relies on several hyperparameters, such as the prior scale, that require careful tuning, which could impact its ease of use and robustness. While the authors acknowledge this issue and discuss the need for a more principled method for prior design, further improvements could be explored. Specifically, I suggest that the authors discuss any sensitivity analyses they performed to understand the impact of these hyperparameters on model performance. This could provide insights into the stability and reliability of the model across a range of settings. Additionally, the authors might consider implementing automated hyperparameter tuning methods, such as Bayesian optimization or hyperparameter sweeps, to alleviate the need for manual tuning. Such techniques could enhance the model's usability and reduce the dependency on empirical parameter selection. Finally, exploring the integration of adaptive or learnable priors might offer a more dynamic approach, allowing the model to adjust parameters in response to varying data conditions automatically.

Some of the modifications to the PaiNN [8] architecture, such as the integration of equivariant flow fields and the use of a Maxwell-Boltzmann prior distribution, appear to be necessary adjustments rather than novel contributions. These changes are essential to adapt PaiNN into a flow-based model while preserving physical symmetries and ensuring accurate molecular dynamics simulations. Additionally, given that similar flow matching techniques have been applied in related work (as discussed in previous papers, such as [9, 10, 11]), the degree of novelty in these architectural modifications may be limited, as they are fundamental requirements for the chosen modeling approach rather than unique innovations. Furthermore, these modifications likely contribute to increased computational costs compared to the original PaiNN architecture, which may limit the model’s efficiency and scalability in large-scale simulations, even if performance is improved for the experiments at hand.

The paper's readability could be significantly improved, as the current structure and presentation make it challenging to follow. Enhancing the flow of the text and providing clearer explanations would greatly benefit readers and better convey the complexity of the work. Location of Table 1 and Table 2 distrupts the flow. Specifically, I suggest reorganizing the sections for better coherence. Large-scale inference and computational costs can be subsection intead of bolds. Additionally, the discussions on the effect of hyperparameters like P and C, as well as the choice of prior, could be moved to the Ablation Studies section following the main experimental results. Each dataset and its results given as 4.2 and 4.3 but it is very hard to pinpoint the results. I think 'effect of prior study' and 'case study'  is where you share results for 4.2. These can be created into a subsetion as 'Results for Universal Model' or something. 'Reproducing kinetic properties' and 'Reproducing structural features' is where you share your results for 4.3, which can also benefit from similar fashion. Figure 2 and Figure 3 can be merge into one figure which spans 2 columns. A figure of the proposed model can be shared in addition to the scheme. I believe these or similar changes would streamline the narrative and make it easier for readers to understand the work and its effect. Initial sentece in the introduction needs a reference as well as the 'Crystalline materials and representation' subsection. 'REPRODUCIBILITY STATEMENT' is given in 11th page.

[1] Liao, Y. L., & Smidt, T. (2022). Equiformer: Equivariant graph attention transformer for 3d atomistic graphs. arXiv preprint arXiv:2206.11990.

[2] Batzner, S., Musaelian, A., Sun, L., Geiger, M., Mailoa, J. P., Kornbluth, M., ... & Kozinsky, B. (2022). E (3)-equivariant graph neural networks for data-efficient and accurate interatomic potentials. Nature communications, 13(1), 2453.

[3] Wu, F., & Li, S. Z. (2023, June). DIFFMD: a geometric diffusion model for molecular dynamics simulations. In Proceedings of the AAAI Conference on Artificial Intelligence (Vol. 37, No. 4, pp. 5321-5329).

[4] Wang, W., & Gómez-Bombarelli, R. (2019). Coarse-graining auto-encoders for molecular dynamics. npj Computational Materials, 5(1), 125.

[5] Tamagnone, S., Laio, A., & Gabrié, M. (2024). Coarse-Grained Molecular Dynamics with Normalizing Flows. Journal of Chemical Theory and Computation, 20(18), 7796-7805.

[6] Chmiela, S., Tkatchenko, A., Sauceda, H. E., Poltavsky, I., Schütt, K. T., & Müller, K. R. (2017). Machine learning of accurate energy-conserving molecular force fields. Science advances, 3(5), e1603015.

[7] Chanussot, L., Das, A., Goyal, S., Lavril, T., Shuaibi, M., Riviere, M., ... & Ulissi, Z. (2021). Open catalyst 2020 (OC20) dataset and community challenges. Acs Catalysis, 11(10), 6059-6072.

[8] Schütt, K., Unke, O., & Gastegger, M. (2021, July). Equivariant message passing for the prediction of tensorial properties and molecular spectra. In International Conference on Machine Learning (pp. 9377-9388). PMLR.

[9] Song, Y., Gong, J., Xu, M., Cao, Z., Lan, Y., Ermon, S., ... & Ma, W. Y. (2024). Equivariant flow matching with hybrid probability transport for 3d molecule generation. Advances in Neural Information Processing Systems, 36.

[10] Klein, L., Krämer, A., & Noé, F. (2024). Equivariant flow matching. Advances in Neural Information Processing Systems, 36.

[11] Dunn, I., & Koes, D. R. (2024). Mixed Continuous and Categorical Flow Matching for 3D De Novo Molecule Generation. ArXiv.

**Questions:**

1) Why authors did not include comparisons with other ML-based MD acceleration methods, such as GNNs, diffusion, variational autoencoders, and normalizing flows. For a fair effectiveness of the model, some of the state-of-the-art ML approaches could have been implemented [1, 2, 3, 4, 5].

2) Can LIFLOW be easily adapted to simulate other materials beyond lithium-based SSEs? If so, what modifications would be necessary? For example, can you apply this model to MD17 [6], OC20 [7], or Ani-1x [8] dataset and show its effectiveness compared to other state-of-the-art?

3) The model relies on several hyperparameters, such as the prior scale, that require careful tuning, potentially impacting its ease of use and robustness. While you acknowledge the need for a more principled method for prior design, did you perform any sensitivity analyses to understand how these hyperparameters affect model performance and stability? Additionally, have you considered implementing automated hyperparameter tuning techniques, like Bayesian optimization or hyperparameter sweeps, to minimize the reliance on manual tuning? Finally, do you see potential benefits in exploring adaptive or learnable priors that could dynamically adjust based on data conditions, and if so, how might this improve the model’s performance and generalizability?

4) Authors mention the risk of generating physically fictitious dynamics. Are there ways to quantify or mitigate this risk more systematically?

5) Amorphous materials often exhibit complex atomic transport mechanisms due to their lack of long-range order [9]. How well would LIFLOW generalize to amorphous systems, and have you considered testing the model on amorphous materials such as amorphous silicon or lithium-phosphorus oxynitride (LiPON) electrolytes [10]? If not, what challenges do you foresee in applying your approach to such systems, and how might the model be adapted to handle the inherent structural disorder?

6) In Section 3.2.1, you provide a comprehensive explanation of how your model ensures invariance to various symmetries, including permutation, translation, and rotation. Given the complexities involved in modeling these symmetries, did you encounter any specific challenges or limitations when implementing these equivariant properties, particularly for higher-order interactions or rare configurations? Additionally, do you believe that incorporating more advanced equivariant architectures (e.g., equivariant graph attention [1]) could further improve the performance or generalizability of your approach?

7) The modifications you made to the PaiNN architecture,  such as integrating equivariant flow fields and using a Maxwell-Boltzmann prior, seem necessary for adapting PaiNN into a flow-based framework that preserves physical symmetries. Given that these changes are essential for the model's operation and that flow matching techniques have been previously applied in related contexts (as seen in [11, 12, 13]), do you consider your approach to be a novel contribution, or do you view it as an adaptation of existing methods for this specific application? Furthermore, how do these modifications impact the computational cost compared to the original PaiNN architecture? Does the accuracy gained with the changes still overshadows the computational cost for the larger simulations?


[1] Liao, Y. L., & Smidt, T. (2022). Equiformer: Equivariant graph attention transformer for 3d atomistic graphs. arXiv preprint arXiv:2206.11990.

[2] Batzner, S., Musaelian, A., Sun, L., Geiger, M., Mailoa, J. P., Kornbluth, M., ... & Kozinsky, B. (2022). E (3)-equivariant graph neural networks for data-efficient and accurate interatomic potentials. Nature communications, 13(1), 2453.

[3] Wu, F., & Li, S. Z. (2023, June). DIFFMD: a geometric diffusion model for molecular dynamics simulations. In Proceedings of the AAAI Conference on Artificial Intelligence (Vol. 37, No. 4, pp. 5321-5329).

[4] Wang, W., & Gómez-Bombarelli, R. (2019). Coarse-graining auto-encoders for molecular dynamics. npj Computational Materials, 5(1), 125.

[5] Tamagnone, S., Laio, A., & Gabrié, M. (2024). Coarse-Grained Molecular Dynamics with Normalizing Flows. Journal of Chemical Theory and Computation, 20(18), 7796-7805.

[6] Chmiela, S., Tkatchenko, A., Sauceda, H. E., Poltavsky, I., Schütt, K. T., & Müller, K. R. (2017). Machine learning of accurate energy-conserving molecular force fields. Science advances, 3(5), e1603015.

[7] Chanussot, L., Das, A., Goyal, S., Lavril, T., Shuaibi, M., Riviere, M., ... & Ulissi, Z. (2021). Open catalyst 2020 (OC20) dataset and community challenges. Acs Catalysis, 11(10), 6059-6072.

[8] Smith, J. S., Zubatyuk, R., Nebgen, B., Lubbers, N., Barros, K., Roitberg, A. E., ... & Tretiak, S. (2020). The ANI-1ccx and ANI-1x data sets, coupled-cluster and density functional theory properties for molecules. Scientific data, 7(1), 134.

[9] Philibert, J. (1991). Atom movements: diffusion and mass transport in solids (p. 577). Les Ulis, France: éditions de Physique.

[10] Lacivita, V., Artrith, N., & Ceder, G. (2018). Structural and compositional factors that control the Li-ion conductivity in LiPON electrolytes. Chemistry of Materials, 30(20), 7077-7090.

[11] Song, Y., Gong, J., Xu, M., Cao, Z., Lan, Y., Ermon, S., ... & Ma, W. Y. (2024). Equivariant flow matching with hybrid probability transport for 3d molecule generation. Advances in Neural Information Processing Systems, 36.

[12] Klein, L., Krämer, A., & Noé, F. (2024). Equivariant flow matching. Advances in Neural Information Processing Systems, 36.

[13] Dunn, I., & Koes, D. R. (2024). Mixed Continuous and Categorical Flow Matching for 3D De Novo Molecule Generation. ArXiv.

---

> ### Author Response · Authors · 2024-11-23
> **Author comments [1/6]**
>
> Dear Reviewer Z7DJ,
>
> First of all, we acknowledge your time and effort in thoroughly reviewing our work, and we appreciate your valuable feedback. Also, thank you for your positive feedback on addressing a relevant challenge in materials research and the transformative potential of our approach. We acknowledge the concerns regarding the presentation of the experimental section and have made significant revisions to the Results section to enhance readability, incorporating most of your suggestions. Additionally, we have updated the manuscript to include more detailed dataset statistics and attached Supplementary Material with additional results for amorphous systems. Please find our point-by-point response below, and we apologize for any potential errors in our reply due to the limited time available for this rebuttal.
>
> ---
>
> ## Weaknesses
>
> > **W1:** The primary comparison is with AIMD simulations and MLIP, which are used to generate the data. The paper does not compare LIFLOW to existing machine learning-based approaches for MD acceleration, such as GNN-based models [1, 2], diffusion models [3], variational approaches [4] or normalizing flows [5]. Inclusion of such comparisons would greatly improve the credibility of the claims of ths work.
>
> Equiformer [1] and NequIP [2] propose equivariant graph neural network architectures for MLIPs. As a ML force field models, they indeed "accelerate" the simulation by approximating the DFT energies and forces, but they still rely on discretizing equations of motion and do not "accelerate" the propagation of dynamics as in our approach. For the universal dataset, both models do not have a trained universal parameters and thus cannot be compared, whereas the universal version of MACE [14] model serves as our reference dynamics. For AIMD models, we compare with the simulation time of MLIP-based simulations in Table 3 (Prediction speed).
>
> DiffMD [3] models short-time dynamics propagation via diffusion models aimed at nearly deterministic dynamics over very small time steps (order of fs), evaluated through accumulated RMSE of snapshots. Consequently, it is not an "acceleration" model for MD simulations, but a surrogate model for real-time dynamics itself.
>
> [4] focuses focuses on learning all-atom to coarse-grained (CG) mappings and CG potentials, a task distinct from learning a propagator. Our work does not aim for exhaustive comparisons among sampling methods, and crystalline systems studied here lack universal CG representations due to mutable bonding and time-variant chemical entities.
>
> [5] describes using a data-driven collective variable (CV) iteratively trained for step proposals, accepted or rejected subsequently. This approach is unsuitable for atomic transport applications considered in our work, because it reduces propagation degrees of freedom to CV space, which is challenging for arbitrary crystalline systems. Identifying optimal CVs is a separate research question. Additionally, the scheme in this work does not guarantee conservation of kinetic properties, which is a key goal of our work.
>
> > **W2:** While the model is shown to work well on lithium SSEs, there is limited discussion on generalizing LIFLOW to other types of materials or chemical systems. This could restrict the perceived impact of the method. An additional showcase on different datasets, such as MD17 [6] and OC20 [7] could further shine light on the model's generalization.
>
> Our primary experiments focus on lithium SSEs because they represent a class of materials with prominent ionic transport and practical kinetic properties. This choice is also influenced by the lack of comprehensive dynamics datasets for material systems.
>
> Regarding the mentioned datasets, MD17 [6] is an AIMD trajectory dataset for small molecules with fine-grained frames over limited time scales. While suitable for benchmarking surrogate models for real-time, nearly-deterministic dynamics, it is much less useful for evaluating ML surrogates in accelerated MD simulations. OC20 [7] provides single-frame structures from DFT relaxation trajectories of catalyst-adsorbate systems, and ANI-1x [8] (from **Q2**) comprises DFT calculations for small molecule structures. Neither dataset includes time series information and cannot benchmark dynamics models.
>
> Relevant works on biomolecular simulations often use MD trajectories for short peptides and fast-folding proteins. While our method is theoretically applicable to any atomistic system, extending it to these simulations would benefit from additional modifications. For example, we did not assume any predefined bonds to generalize across materials with varying compositions, but incorporating unbreakable bonds as inductive biases for biomolecules under physiological conditions would improve accuracy.

---

> ### Author Response · Authors · 2024-11-23
> **Author comments [2/6]**
>
> > **W3:** The model struggles with extrapolating to conditions outside the training regime, such as lower temperatures, where rare events are poorly captured. This suggests potential limitations in scenarios requiring greater generalization. Considering well know enhancement approaches, such as physics informed neural networks for the loss function modification or active learning to sample when rare events occur can be benificial.
>
> We thank the reviewer for the valuable suggestions. We acknowledge that dealing with data scarcity and model generalization are crucial topics in the relevant literature. The recommended approaches, such as Physics-Informed Neural Networks and active learning, are indeed effective strategies for enhancing generalization. Our use of an adaptive prior and a symmetry-aware model architecture aligns with this goal, representing physics-inspired efforts to improve model generalization. The incorporation of techniques like active learning is a promising future direction, as such methods can be developed orthogonally to our current approach. Additionally, we would like to emphasize that our method successfully extrapolated to materials with different compositions (Sec. 4.2) and to extended temporal and spatial scales (Sec. 4.3). While additional training data would further improve generalization, our primary objective was to illustrate the model's inherent generalization ability, rather than to develop a foundation model capable of generalizing universally.
>
> > **W4:** The model relies on several hyperparameters, such as the prior scale, that require careful tuning, which could impact its ease of use and robustness. While the authors acknowledge this issue and discuss the need for a more principled method for prior design, further improvements could be explored. Specifically, I suggest that the authors discuss any sensitivity analyses they performed to understand the impact of these hyperparameters on model performance. This could provide insights into the stability and reliability of the model across a range of settings. Additionally, the authors might consider implementing automated hyperparameter tuning methods, such as Bayesian optimization or hyperparameter sweeps, to alleviate the need for manual tuning. Such techniques could enhance the model's usability and reduce the dependency on empirical parameter selection. Finally, exploring the integration of adaptive or learnable priors might offer a more dynamic approach, allowing the model to adjust parameters in response to varying data conditions automatically.
>
> We acknowledge the reviewer for pointing this out, and we have conducted additional ablation studies on hyperparameter sensitivity, which is gathered in Appendix E.3 in the revised manuscript. Using the LGPS dataset experiment as an example, we evaluated the impact of the Propagator prior scales (lithium and frame) and the Corrector noise scales, varying each scale from x1/2 to x2 and measuring the resulting diffusivity from each trained model. We found that diffusivity values show minor deviations from their peak value at the optimal Propagator scales, and changing Corrector noise scales larger than a certain threshold causes diffusivities to decrease, suggesting that stronger correction enhances stability at a cost of slightly diminished diffusive behavior.
>
> Additionally, we performed ablation studies on the Corrector model. Compared to the universal model, where the Corrector is highly beneficial (Table 1), training and inference on AIMD models are conducted on materials with the same composition, resulting in less pronounced Propagator errors. Additional experiments examined simplifying Corrector inference by reducing the number of flow integration steps or applying the Corrector intermittently across Propagator steps for LGPS AIMD models (Appendix E.3, Fig. A8). Results showed that while these simplifications do not significantly affect diffusivity values in 150-step LGPS inference, the Corrector remains critical for extending dynamics up to 1,000-step inference at higher temperatures.
>
> Regarding additional suggestions from the reviewer, we note that Bayesian optimization and hyperparameter sweeps are certainly applicable, as optimal hyperparameters are identifiable according to our ablation studies. However, these are more technical aspects requiring a larger computational budget, so we defer systematic studies of such methods to future work. Adaptive or learnable priors would indeed be beneficial. Our introduction of adaptive scaling for the Maxwell-Boltzmann prior already partially incorporates such a mechanism, demonstrating empirical advantages. Extending this to make the prior fully learnable from flow-matching training is left for future exploration. We would greatly appreciate literature recommendations from the reviewer on relevant learnable priors for generative modeling.

---

> ### Author Response · Authors · 2024-11-23
> **Author comments [3/6]**
>
> > **W5:** Some of the modifications to the PaiNN [8] architecture, such as the integration of equivariant flow fields and the use of a Maxwell-Boltzmann prior distribution, appear to be necessary adjustments rather than novel contributions. These changes are essential to adapt PaiNN into a flow-based model while preserving physical symmetries and ensuring accurate molecular dynamics simulations. Additionally, given that similar flow matching techniques have been applied in related work (as discussed in previous papers, such as [9, 10, 11]), the degree of novelty in these architectural modifications may be limited, as they are fundamental requirements for the chosen modeling approach rather than unique innovations. Furthermore, these modifications likely contribute to increased computational costs compared to the original PaiNN architecture, which may limit the model’s efficiency and scalability in large-scale simulations, even if performance is improved for the experiments at hand.
>
> First of all, we want to clarify that use of Maxwell-Boltzmann prior is pertinent to the "prior" part of generative modeling, which is totally separate from the "equivariant flow fields" modeled by PaiNN [8] architecture. As could be seen in the ablation studies (Table 1, Exp 1), the use of Maxwell-Boltzmann prior is indeed a novel contribution of our work, which is not a necessary adjustment because a normal distribution could still be used with degraded performance.
>
> We note that the modifications to the PaiNN architecture made here, would be not unique in a sense that similar approaches to incorporate two structures has been applied in previous works [15, 16], but are not necessary adjustments to make it usable as an equivariant flow fields. This is different from previous works on equivariant flows [9, 10, 11] mentioned by the reviewer such that either two set of structures or structure + displacement is required as inputs. While the original PaiNN model is theoretically capable of handling displacement inputs (vector input features), the modifications in the architecture could be understood as empirical adjustment resulting from investigating how the incorporating intermediate displacement are best utilized in our set of experiments.
>
> Finally, the modification essentially involves maintaining two copies of edge information instead of one in the original graph, ensuring that linear scaling is preserved even as the simulation scale increases. For instance, in Table 3, a 200-atom cell requires 48 seconds, while a 3,200-atom cell takes 352 seconds for 1,000-step inference—a growth rate that is slightly sublinear, likely due to parallelization. Thus, we confirm that the modification does not compromise scalability at larger scales.
>
> > **W6:** The paper's readability could be significantly improved, as the current structure and presentation make it challenging to follow. Enhancing the flow of the text and providing clearer explanations would greatly benefit readers and better convey the complexity of the work. Location of Table 1 and Table 2 distrupts the flow. Specifically, I suggest reorganizing the sections for better coherence. Large-scale inference and computational costs can be subsection intead of bolds. Additionally, the discussions on the effect of hyperparameters like P and C, as well as the choice of prior, could be moved to the Ablation Studies section following the main experimental results. Each dataset and its results given as 4.2 and 4.3 but it is very hard to pinpoint the results. I think 'effect of prior study' and 'case study' is where you share results for 4.2. These can be created into a subsetion as 'Results for Universal Model' or something. 'Reproducing kinetic properties' and 'Reproducing structural features' is where you share your results for 4.3, which can also benefit from similar fashion. Figure 2 and Figure 3 can be merge into one figure which spans 2 columns. A figure of the proposed model can be shared in addition to the scheme. I believe these or similar changes would streamline the narrative and make it easier for readers to understand the work and its effect. Initial sentece in the introduction needs a reference as well as the 'Crystalline materials and representation' subsection. 'REPRODUCIBILITY STATEMENT' is given in 11th page.
>
> We would like to thank you for your thoughtful comments and suggestions, which have greatly helped improve the clarity and flow of the manuscript. In response to the points raised, we have revised the entire Experiments section (Sec. 4) to streamline the presentation of results. We ensured that key results are presented first, followed by ablation studies and less critical details. We merged relevant tables and figures as suggested and placed them near the corresponding part in the main text. Point-by-point replies are as follows:

---

> ### Author Response · Authors · 2024-11-23
> **Author comments [4/6]**
>
> * Table 1 and Table 2: We have consolidated these into a single table (Table 1) with subheadings for each ablation study, improving coherence and ease of reference. Additionally, we have added a color scheme to Table 1 to enhance the visual interpretability of the results.
> * Large-scale inference and computational costs: As this section pertains to AIMD models (specifically LGPS), we have kept it as originally written. However, Computational Costs has been restructured into a separate subsection (Sec. 4.4) for better organization.
> * Effects of hyperparameters and choice of prior: We have moved the discussions on the effects of hyperparameters and Corrector, as well as the choice of Propagator prior, to the Ablation Studies subsection (Sec. 4.2.2), following the main experimental results (Sec. 4.2.1), as suggested. Additionally, we extracted the key elements from Fig. A4 and created a new figure (Fig. 2) in the main text, under the paragraph “Reproducing kinetic properties” in Sec. 4.2.1, to ensure the discussion in the main text remains self-contained.
> * “Reproducing kinetic properties” and “Reproducing structural features”: Since Sec. 4.3 primarily presents results, we have left it unchanged. We believe this structure is appropriate and aligns with the goal of showcasing the results clearly.
> * Fig. 2 and Fig. 3: We have merged these figures into a single figure spanning the text width (Fig. 3 in the revised manuscript), as per your suggestion.
> * A figure of the proposed model: The PaiNN model used in this work is adopted with a small modification. Since the existing scheme sufficiently conveys the model’s structure, we believe no additional figure is necessary and defer the details of model architecture to the Appendix D.2.
> * Extra citations: For the initial sentence in the Introduction, we have added references to a general introductory textbook on kinetics and atomic transport in materials [17] and a general introduction to atomistic simulations in materials science [18]. For “Crystalline materials and representation” paragraph in Sec. 2.1, since the latter part of representation already has a citation, we have added a reference regarding the periodic representation of the crystal structure [19].
> * “REPRODUCIBILITY STATEMENT”: While our Reproducibility Statement is on the 11th page, we consider it aligned with the ICLR guidelines: “This optional reproducibility statement will not count toward the page limit, but should not be more than 1 page.” (copied verbatim from the ICLR Author Guide).
>
> We hope these revisions address the your concerns and improve the manuscript's overall readability and organization. Thank you again for your valuable feedback.
>
> ## Questions
>
> > **Q1:** Why authors did not include comparisons with other ML-based MD acceleration methods, such as GNNs, diffusion, variational autoencoders, and normalizing flows. For a fair effectiveness of the model, some of the state-of-the-art ML approaches could have been implemented [1, 2, 3, 4, 5].
>
> Please refer to our comments in **W1** (in Author comments 1).
>
> > **Q2:** Can LIFLOW be easily adapted to simulate other materials beyond lithium-based SSEs? If so, what modifications would be necessary? For example, can you apply this model to MD17 [6], OC20 [7], or Ani-1x [8] dataset and show its effectiveness compared to other state-of-the-art?
>
> Please refer to our comments in **W2** (in Author comments 1).
>
> > **Q3:** The model relies on several hyperparameters, such as the prior scale, that require careful tuning, potentially impacting its ease of use and robustness. While you acknowledge the need for a more principled method for prior design, did you perform any sensitivity analyses to understand how these hyperparameters affect model performance and stability? Additionally, have you considered implementing automated hyperparameter tuning techniques, like Bayesian optimization or hyperparameter sweeps, to minimize the reliance on manual tuning? Finally, do you see potential benefits in exploring adaptive or learnable priors that could dynamically adjust based on data conditions, and if so, how might this improve the model’s performance and generalizability?
>
> Please refer to our comments in **W4** (in Author comments 2).
>
> > **Q4:** Authors mention the risk of generating physically fictitious dynamics. Are there ways to quantify or mitigate this risk more systematically?
>
> Our intention when mentioning the number of numerically stable steps and generation of physically fictitious dynamics was that numerical stability of propagation is not a perfect measure of “stability” in a physical sense, so it requires evaluation alongside kinetic and structural metrics, such as log MSD and RDF. Physically fictitious behavior is reflected in poorer kinetic metrics, as illustrated by example trajectories in Fig. A4. We have clarified our discussion for this part to convey our intention clearer.

---

> ### Author Response · Authors · 2024-11-23
> **Author comments [5/6]**
>
> As shown in Fig. A4II, we noticed that hydrogens often show fictitious diffusive behavior, due to their light mass leading to large prior displacements under Maxwell-Boltzmann distribution. Propagator sometimes struggle to reconcile these displacements with smaller observed displacements. Interestingly, this is a challenge also encountered in classical/ab initio MD, where light mass of hydrogen atoms are problematic when using longer time steps. Hence, one could introduce constraints to hydrogens, similarly MD simulations, or treat hydrogens differently, as we did for lithium in electrolyte simulations.
>
> Another way of mitigating this could leverage explicitly incorporating pre-trained representations or physical information. A concurrent ICLR submission, “Boltzmann priors for Implicit Transfer Operators” [20], demonstrates that interpolating equilibrium models (e.g., pre-trained Boltzmann generators) with transfer operator models enhances propagation stability and data efficiency. We anticipate similar strategies could mitigate fictitious dynamics in materials systems, given the limited availability of high-quality dynamics data.
>
> > **Q5:** Amorphous materials often exhibit complex atomic transport mechanisms due to their lack of long-range order [9]. How well would LIFLOW generalize to amorphous systems, and have you considered testing the model on amorphous materials such as amorphous silicon or lithium-phosphorus oxynitride (LiPON) electrolytes [10]? If not, what challenges do you foresee in applying your approach to such systems, and how might the model be adapted to handle the inherent structural disorder?
>
> We conducted a preliminary benchmark during the rebuttal period using the amorphous lithium phosphate structure and trajectory data from [21]. The results, included in the Supplementary Material (summary slide and trajectory videos), indicate that the LiFlow methodology is applicable to amorphous systems with reasonable accuracy for both kinetic and structural observables. Interestingly, the diversity of atomic configurations in amorphous systems slowed the convergence of the Propagator model during training, taking approximately four times longer than for crystalline systems.
>
> > **Q6:** In Section 3.2.1, you provide a comprehensive explanation of how your model ensures invariance to various symmetries, including permutation, translation, and rotation. Given the complexities involved in modeling these symmetries, did you encounter any specific challenges or limitations when implementing these equivariant properties, particularly for higher-order interactions or rare configurations? Additionally, do you believe that incorporating more advanced equivariant architectures (e.g., equivariant graph attention [1]) could further improve the performance or generalizability of your approach?
>
> While the main complexities are relevant to the task design, the symmetries are ensured by the representation of materials systems and simple modifications to the PaiNN model. As the reviewer noted, since PaiNN utilizes features up to L = 1, higher-order interactions are not fully captured in the flow model architecture in this work. The designed task and symmetry requirements allow the use of any architecture capable of processing two sets of positional information (L = 1) and outputting a vector field for displacements (L = 1). We selected PaiNN for its fast inference speed and strong benchmarking performance in ML interatomic potential literature. While an ablation study on different message-passing architectures would be interesting, it is outside the scope of this study. The Equiformer [1] model mentioned by reviewer, has been updated with modifications from SO(3) convolutions with eSCN [22] convolution (eqV2, [23]), and has demonstrated excellent scalability in OMat24 dataset [24]. While it could enhance generalization with larger datasets and computational resources, its use in dynamics simulations is currently impractical due to high computational costs. We plan to explore alternative architectural choices in future work.
>
> > **Q7:** The modifications you made to the PaiNN architecture, such as integrating equivariant flow fields and using a Maxwell-Boltzmann prior, seem necessary for adapting PaiNN into a flow-based framework that preserves physical symmetries. Given that these changes are essential for the model's operation and that flow matching techniques have been previously applied in related contexts (as seen in [11, 12, 13]), do you consider your approach to be a novel contribution, or do you view it as an adaptation of existing methods for this specific application? Furthermore, how do these modifications impact the computational cost compared to the original PaiNN architecture? Does the accuracy gained with the changes still overshadows the computational cost for the larger simulations?
>
> Please refer to our comments in **W5** (in Author comments 3).

---

> > ### Author Response · Authors · 2024-11-23
> > **Author comments [6/6]**
> >
> > ### References
> > References [1–13] are adapted from the Reviewer's original comment.
> >
> > [14] Batatia et al., A foundation model for atomistic materials chemistry, 2024. URL https://arxiv.org/abs/2401.00096
> >
> > [15] Schreiner et al., Implicit transfer operator learning: Multiple time-resolution models for molecular dynamics. Advances in Neural Information Processing Systems, volume 36, pp. 36449–36462, 2023.
> >
> > [16] Hsu et al., Score dynamics: Scaling molecular dynamics with picoseconds time steps via conditional diffusion model. J. Chem. Theory Comput., 20(6):2335–2348, 2024.
> >
> > [17] Balluffi et al., Kinetics of Materials. John Wiley & Sons, 2005.
> >
> > [18] Sidney Yip, Molecular Mechanisms in Materials: Insights from Atomistic Modeling and Simulation. MIT Press, 2023.
> >
> > [19] Ashcroft and Mermin. Solid State Physics. Saunders College Publishing, 1976.
> >
> > [20] Anonymous, Boltzmann priors for Implicit Transfer Operators, Submitted to The Thirteenth International Conference on Learning Representations, 2024. URL https://openreview.net/forum?id=pRCOZllZdT
> >
> > [21] Jun et al., The nonexistence of a paddlewheel effect in superionic conductors. Proc. Natl. Acad. Sci. U.S.A., 121(18):e2316493121, 2024.
> >
> > [22] Passaro and Zitnick, Reducing SO(3) Convolutions to SO(2) for Efficient Equivariant GNNs, Proceedings of the 40th International Conference on Machine Learning, volume 202, pp. 27420–27438, 2023.
> >
> > [23] Liao et al., EquiformerV2: Improved Equivariant Transformer for Scaling to Higher-Degree Representations, In The Twelfth International Conference on Learning Representations, 2024.
> >
> > [24] Barroso-Luque et al., Open Materials 2024 (OMat24) Inorganic Materials Dataset and Models, 2024. URL https://arxiv.org/abs/2410.12771
> >
> > ---
> > Thank you again for your time and effort in reviewing our work. We would also appreciate it if you could reconsider the scores, should you feel that our revisions and responses have addressed your concerns and merit a more favorable evaluation. Please let us know if you have any further suggestions or questions.
> >
> > Best regards,
> >
> > Submission11944 Authors

---

> > > ### Author Response · Authors · 2024-11-25
> > > **Friendly Reminder for Reviewer Feedback**
> > >
> > > Dear Reviewer Z7DJ,
> > >
> > > We wanted to kindly remind you that the rebuttal period ends tomorrow. We greatly appreciate your time and effort in reviewing our work and would be grateful if you could take a moment to review our rebuttal. Any feedback you can provide would be very helpful, and if you have any remaining concerns, we would be happy to address them.
> > >
> > > Specifically, we’d appreciate it if you could let us know if there are any concerns where you remain **absolutely certain (5)** in your assessment that the manuscript should be **rejected (3)**, particularly in relation to the revised manuscript or our discussion, as this will help us better understand the critical issues with the work.
> > >
> > > Thank you again for your thoughtful consideration.
> > >
> > > Best regards,
> > >
> > > Submission11944 Authors

---

> ### Comment · Reviewer_Z7DJ · 2024-11-25
>
> I would like to thank the authors for their thoughtful responses and the progress highlighted in their rebuttal.
>
> I have revised my ratings based on the updates provided in the rebuttal. While the presentation of the work has improved significantly, I still believe the paper may be better suited for a workshop rather than the main conference. This is due to its limited theoretical contributions to the field and the absence of comprehensive benchmarking.

---

> > ### Author Response · Authors · 2024-11-26
> > **Asking for clarification**
> >
> > Dear Reviewer Z7DJ,
> >
> > Thank you for reviewing our rebuttal. We appreciate the reviewer's acknowledgment of our improved presentation. To clarify the remaining concerns, we hope to address any remaining concerns and demonstrate the suitability of the paper for this venue.
> >
> > We identified the following items from your original review comments corresponding to the raised concerns:
> > - Limited theoretical contributions
> >   - **W5:** Some of the modifications to the PaiNN [8] architecture, ...
> > - Absence of comprehensive benchmark
> >   - **W1:** The primary comparison is with AIMD simulations ...
> >   - **W2:** While the model is shown to work well on lithium SSEs, ...
> >   - **W4:** The model relies on several hyperparameters, ...
> >   - **Q5:** Amorphous materials often exhibit complex atomic transport mechanisms ...
> >
> > Are there specific gaps in our response that contribute to the overall assessment remaining unchanged? With the discussion period extended by six days, we would be happy to engage further to address any remaining concerns with you.
> >
> > Thank you,
> >
> > Submission11944 Authors

---

### Official Review · Reviewer_UKGb · 2024-11-05

**Soundness:** 4
**Presentation:** 4
**Contribution:** 3
**Rating:** 8
**Confidence:** 2

**Summary:**

The LiFlow framework presents an innovative ML-based approach to accelerating molecular dynamics simulations specifically for atomic transport. By combining a propagator and corrector network in a symmetry-aware conditional flow matching framework, the model achieves high performance while maintaining physical consistency across larger time scales and system sizes.

**Strengths:**

- The paper is clearly written, with thorough notation and well-organized sections, making it accessible and valuable to the AI for Science community. Its presentation facilitates understanding of the model's design and methodology.
- The model architecture is built with strong symmetry considerations, effectively leveraging domain-specific invariances essential for applications in atomic transport. This symmetry-aware approach not only improves the model's robustness but also reinforces its applicability to material science challenges.

**Weaknesses:**

- While the model is effective for atomic transport in crystalline materials, its applicability to other systems like biomolecules or amorphous structures remains unexplored. Extending this method to diverse types of molecular dynamics systems could increase its impact.
- While the paper provides dataset statistics of elemental count distribution and Hitogram of MSD, it would benefit from including detailed statistics of the dataset. Information such as the number of atoms, types of atoms, and lattice structures used in the training and testing sets would offer a more comprehensive understanding of the model's training conditions and its generalizability across different material structures.

**Questions:**

- In principle, it seems the propagator network alone should be sufficient for simulating MD if it’s well-designed. Why was the corrector network included, and what specific benefits does it provide? Was there a reason for not focusing on further enhancing the propagator network instead?
- What aspects of this method make it particularly well-suited for atomic transport? Would it also apply to simulating protein MD, such as Alanine Dipeptide?
- Is there a reason for not comparing this approach with other ML-based MD acceleration methods, such as Time-warp?
- In Table A1, the model appears sensitive to the choice of internal GNN architecture and representation. What were the main considerations when selecting the GNN, and is there another option instead of a modified PINN?
- Does the model maintain supercell and global translation invariance within the corrector network? I noticed that in Appendix D.2, edge construction on the denoised positions was skipped—how might this impact these invariances?

**Details Of Ethics Concerns:**

While this paper addresses molecular dynamics, which is highly valuable in chemistry, physics, and materials science, the author does not mention any concerns about the potential misuse of this technology. For instance, there could be risks of malicious applications, such as creating chemical weapons. Although such scenarios are unlikely based on current understanding, I believe all researchers working in AI for scientific applications should remain vigilant about these possibilities, given that AI tools can be used without specialized knowledge.

---

> ### Author Response · Authors · 2024-11-23
> **Author comments [1/3]**
>
> Dear Reviewer UKGb,
>
> Thank you for your positive feedback on the clarity and organization of our work. In response to your suggestions, we have revised the manuscript to include more detailed dataset statistics and have attached Supplementary Material with additional results for amorphous systems. Furthermore, we have improved the exposition of the Results section to enhance readability. Please find our point-by-point response below, and we apologize for any potential errors in our reply due to the limited time available for this rebuttal.
>
> ---
>
> ## Weaknesses
> > **W1:** While the model is effective for atomic transport in crystalline materials, its applicability to other systems like biomolecules or amorphous structures remains unexplored. Extending this method to diverse types of molecular dynamics systems could increase its impact.
>
> Thank you for your thoughtful suggestion. While the methodology developed in this work is, in principle, applicable to arbitrary atomistic systems, some key contributions are specifically tailored to atomic transport in material systems. For example, we proposed using the (scaled) Maxwell-Boltzmann prior, as it represents the marginal distribution of atomic velocities in the short-time limit. Biomolecules are often polymeric with single connected entities, and face sidechain interactions that hinder rapid atomic movement, which introduces entropic barriers to transitions. Consequently, biomolecular simulations typically target much longer timescales than those considered here. Additionally, the unbreakable bonds in biomolecules under physiological conditions could be incorporated as inductive biases for improved accuracy, while we did not assume the existence of bonds a priori to ensure generalization across materials with varying compositions.
>
> Regarding amorphous systems, we conducted a preliminary benchmark during the rebuttal period using the amorphous lithium phosphate structure and trajectory data from [1]. The results, included in the Supplementary Material (summary slide and trajectory videos), indicate that the LiFlow methodology is applicable to amorphous systems with reasonable accuracy for both kinetic and structural observables. Interestingly, the diversity of atomic configurations in amorphous systems slowed the convergence of the Propagator model during training, taking approximately four times longer than for crystalline systems. As suggested, we will continue to explore extending our method to a broader range of atomistic systems.
>
> > **W2:** While the paper provides dataset statistics of elemental count distribution and Hitogram of MSD, it would benefit from including detailed statistics of the dataset. Information such as the number of atoms, types of atoms, and lattice structures used in the training and testing sets would offer a more comprehensive understanding of the model's training conditions and its generalizability across different material structures.
>
> Thank you for your valuable suggestion of adding detailed dataset statistics. In response, we added distributions of atom counts, element counts per structure, and space group distributions to Appendix Fig. A1. To summarize, most structures contain 100–200 atoms (up to ~600, Fig. A1c) and consist of 3–6 elements (up to 8, Fig. A1d). Crystal systems are sampled from a diverse range of existing space groups, with over half of crystals having triclinic or monoclinic lattice systems (Fig. A1e). Given that the training and test sets are split by composition, we believe this further demonstrates the generalizability of our model.
>
> ## Questions
>
> > **Q1:** In principle, it seems the propagator network alone should be sufficient for simulating MD if it’s well-designed. Why was the corrector network included, and what specific benefits does it provide? Was there a reason for not focusing on further enhancing the propagator network instead?
>
> The stochastic nature of the LiFlow Propagator model requires a substantial dataset to sufficiently capture the distribution of potential atomic movements over extended time intervals. For the universal MLIP dataset model, however, data collection through MD simulations using MLIPs across diverse materials structures poses challenges in achieving dataset sizes comparable to those in biomolecular simulations with classical force fields (as in previous methods introduced in Sec. 2.2). This limitation leads to inevitable Propagator prediction errors, further compounded by the autoregressive nature of inference, causing larger divergence over time. The Corrector model mitigates this issue by mapping erroneous atom positions after propagation to physically plausible distributions, thereby stabilizing propagation and enabling longer simulation steps.

---

> ### Author Response · Authors · 2024-11-23
> **Author comments [2/3]**
>
> Note that, when training and inference are performed on materials with the same composition (AIMD models), Propagator errors are less pronounced compared to the universal model, where the Corrector proves highly beneficial (Table 1). Additional experiments explored simplifying Corrector inference by reducing the number of flow integration steps or applying the Corrector intermittently across Propagator steps (Appendix E.3, Fig. A8). Results indicated that while simplifying Corrector inference does not significantly affect diffusivity values in 150-step LGPS inference, the Corrector remains essential for extending dynamics up to 1,000-step inference at higher temperatures.
>
> > **Q2:** What aspects of this method make it particularly well-suited for atomic transport? Would it also apply to simulating protein MD, such as Alanine Dipeptide?
>
> As mentioned in the previous reply, the adaptive Maxwell-Boltzmann prior is motivated from the marginal distribution of atomic velocities in the short-time and non-interacting limit. Biomolecules are often polymeric with single connected entities, and face sidechain interactions that hinder rapid atomic movement, which introduces entropic barriers to transitions. Consequently, biomolecular simulations typically target much longer timescales than those considered in this work. Furthermore, the unbreakable bonds in biomolecules under physiological conditions could be incorporated as inductive biases for improved accuracy, while we did not assume the existence of bonds a priori to ensure generalization across materials with varying compositions.
>
> > **Q3:** Is there a reason for not comparing this approach with other ML-based MD acceleration methods, such as Time-warp?
>
> Our primary contribution lies not in the general use of flow matching or neural ODEs for MD surrogates but in their application to atomic transport in crystalline systems. Previous methods are not directly applicable to our study because the choice of generative modeling approach depends on the specific atomistic system and the target physical property.
>
> For example, Timewarp [2] aims to accelerate the sampling over equilibrium conformational states as measured by effective sample size per simulation time. Hence, their propagator is a normalizing flow model that gives exact likelihood as a proposal distribution which is then accepted or rejected by Metropolis-Hastings algorithm. While this ensures that samples are thermodynamically consistent with the Boltzmann distribution, it does not preserve kinetics of the reference dynamics and most of the proposals are rejected during MCMC. Our main objective is reproducing kinetic observables related to atomic transport with faster kinetics at elevated temperatures and mutable chemical environments. Thus, we opted for (1) a rejection-free Corrector mechanism for improved simulation efficiency and (2) flow matching instead of normalizing flows because we could prioritize expressivity over exact likelihoods.
>
> > **Q4:** In Table A1, the model appears sensitive to the choice of internal GNN architecture and representation. What were the main considerations when selecting the GNN, and is there another option instead of a modified PINN?
>
> The designed task and symmetry requirements allow the use of any architecture capable of processing two sets of positional information (L = 1) and outputting a vector field for displacements (L = 1). We selected PaiNN [3] for its fast inference speed and strong benchmarking performance in ML interatomic potential literature. While an ablation study on different message-passing architectures would be interesting, it is outside the scope of this study.
>
> Additionally, we want to clarify that Table A1 highlights performance differences based on how input features are utilized within the chosen message-passing architecture, rather than a comparison of different message-passing schemes. For modified PaiNN, we initialize node features with current step displacements and compute two distance vectors using both current and displaced coordinates for message passing, as detailed in Appendix D.2. Although this approach may seem redundant, given that the model already receives displacements as inputs, Table A1 shows that explicitly incorporating distances from both coordinate sets significantly improves performance.
>
> > **Q5:** Does the model maintain supercell and global translation invariance within the corrector network? I noticed that in Appendix D.2, edge construction on the denoised positions was skipped—how might this impact these invariances?

---

> ### Author Response · Authors · 2024-11-23
> **Author comments [3/3]**
>
> In the context of the Corrector network, the unlabeled positions $X$ (and $X’$) and displacements $D$ (and $D’$) in global translation invariance (eq. (4)) and supercell invariance (eq. (7)) represent noised positions and denoising displacements, respectively. Therefore, as long as edge construction is based on noised positions, the modified PaiNN model output remains invariant to global translation (due to edge vector invariance) and to supercell transformations (as the local atomic neighborhood is preserved under periodicity) acting on noised positions.
>
> ## Ethics Concerns
> > **Details Of Ethics Concerns:** While this paper addresses molecular dynamics, which is highly valuable in chemistry, physics, and materials science, the author does not mention any concerns about the potential misuse of this technology. For instance, there could be risks of malicious applications, such as creating chemical weapons. Although such scenarios are unlikely based on current understanding, I believe all researchers working in AI for scientific applications should remain vigilant about these possibilities, given that AI tools can be used without specialized knowledge.
>
> We regret neglecting this important ethical consideration in the initial submission and appreciate the reviewer for pointing this out. In response, we have added a comprehensive ethics statement to the paper, which appears after the Conclusion, to address the potential risks and ensure responsible use of our work.
>
> **Ethics Statement**
>
> This work raises ethical considerations related to the general use of machine learning in scientific simulations, particularly in the context of molecular dynamics. While the model presented, LiFlow, is intended to accelerate dynamics simulations for materials science, there is a potential for misuse in harmful applications, such as the development of dangerous materials or chemicals. Although unlikely in the current form of our methodology, we acknowledge the following potential scenarios for misuse, as ML-driven simulations could be misused to design materials with undesirable properties, such as highly reactive compounds that may be hazardous to health or the environment:
>
> * Environmentally harmful materials: Simulations could lead to the creation of materials that, when manufactured or disposed of, could pose long-term environmental risks, such as non-biodegradable or highly polluting compounds.
> * Unstable materials: Inaccurate predictions or malicious use of this framework could result in the generation of materials with undesirable or unstable properties, such as those prone to explosive reactions or dangerous degradation.
> * Chemical weapons: Simulations may be applied to develop advanced nanomaterials with toxicological risks or harmful capabilities, including those used in biological or chemical warfare.
>
> To mitigate these risks, we commit to working closely with materials experts to ensure responsible usage and oversight of the methodological developments. Additionally, no human subjects, sensitive data, or privacy-related issues are involved in this study, and there are no conflicts of interest or external sponsorships associated with this work.
>
> ### References
> [1] Jun et al., The nonexistence of a paddlewheel effect in superionic conductors. Proc. Natl. Acad. Sci. U.S.A., 121(18):e2316493121, 2024.
>
> [2] Klein et al., Timewarp: Transferable acceleration of molecular dynamics by learning time-coarsened dynamics. Advances in Neural Information Processing Systems, volume 36, pp. 52863–52883, 2023.
>
> [3] Schütt et al., Equivariant message passing for the prediction of tensorial properties and molecular spectra. Proceedings of the 38th International Conference on Machine Learning, volume 139, pp. 9377–9388, 2021.
>
> ---
> Thank you again for your time and effort in reviewing our work, and please let us know if you have any further suggestions or questions.
>
> Best regards,
>
> Submission11944 Authors

---

> > ### Comment · Reviewer_UKGb · 2024-11-26
> >
> > Thank you and my concerns are resolved. I will stand by my score.

---

> > > ### Author Response · Authors · 2024-11-26
> > > **Thank you**
> > >
> > > Dear Reviewer UKGb,
> > >
> > > Thank you for reviewing our comments and confirming that the concerns have been resolved. We greatly appreciate your positive feedback and your attention to the ethics concerns.
> > >
> > > Thank you,
> > >
> > > Submission11944 Authors

---

### Meta-Review · Area_Chair_WjjX · 2024-12-19

**Metareview:**

The manuscript presents LiFlow, a generative framework designed to accelerate molecular dynamics (MD) simulations for crystalline materials, particularly lithium diffusion in solid-state electrolytes (SSEs). The proposed approach leverages flow matching with two key components: a Propagator for predicting atomic displacements and a Corrector for maintaining physical plausibility. These are integrated into a symmetry-aware architecture that incorporates an adaptive Maxwell–Boltzmann prior to account for chemical and thermal conditions.

While the technical implementation and the empirical results are compelling, the paper has certain limitations. Several of the architectural modifications, such as the use of equivariant flow fields and the Maxwell-Boltzmann prior, are essential adaptations of existing frameworks rather than novel innovations. Additionally, the overarching framework of flow-based generative modeling for MD surrogate tasks has been explored in prior work though in different application contexts. Specific elements, such as the predictor-corrector approach, the use of two sets of edges in conditional GNNs, and the incorporation of periodicity for crystalline systems, are grounded in well-established methods or previously published studies.

After careful consideration, the reviewers recommended that the manuscript not be accepted for publication in its current form. While the study is technically sound and demonstrates promising results, the level of conceptual novelty was judged not significant enough for ICLR. That said, the approach highlights an important research direction, and with further development, including more innovative methodological contributions and stronger practical validations, this work could become a valuable addition to the literature. The authors are encouraged to build on the solid foundation presented here and pursue these improvements for future submissions.

**Additional Comments On Reviewer Discussion:**

During the rebuttal period, the authors provided additional simulations and clarifications that addressed some of the concerns raised during the initial review (eg. sensitivity to scales provided by the users). These comparisons helped to highlight the distinct motivations and applications of LiFlow, particularly its focus on crystalline materials and lithium diffusion, which differ from the broader applications of competing methods in protein dynamics or general molecular systems. The authors also clarified that certain architectural modifications, while appearing incremental, were carefully designed to respect the unique symmetries and constraints of crystalline systems. These explanations provided greater context for the methodological choices and helped to better situate the work within the landscape of molecular dynamics surrogate modeling.

---

### Decision · Program_Chairs · 2025-01-22

Reject